# No-Regret Learning Dynamics for Extensive-Form Correlated Equilibrium

**Andrea Celli**[*]
Politecnico di Milano
andrea.celli@polimi.it

**Alberto Marchesi**[*]
Politecnico di Milano
alberto.marchesi@polimi.it

**Gabriele Farina**[*]
Carnegie Mellon University
gfarina@cs.cmu.edu

**Nicola Gatti**
Politecnico di Milano
nicola.gatti@polimi.it

## Abstract

The existence of simple, uncoupled no-regret dynamics that converge to correlated equilibria in normal-form games is a celebrated result in the theory of multi-agent systems. Specifically, it has been known for more than 20 years that when all players seek to minimize their *internal* regret in a repeated normal-form game, the empirical frequency of play converges to a normal-form correlated equilibrium. Extensive-form (that is, tree-form) games generalize normal-form games by modeling both sequential and simultaneous moves, as well as private information. Because of the sequential nature and presence of partial information in the game, extensive-form correlation has significantly different properties than the normal-form counterpart, many of which are still open research directions. Extensive-form correlated equilibrium (EFCE) has been proposed as the natural extensive-form counterpart to normal-form correlated equilibrium. However, it was currently unknown whether EFCE emerges as the result of uncoupled agent dynamics. In this paper, we give the first uncoupled no-regret dynamics that converge to the set of EFCEs in $n$-player general-sum extensive-form games with perfect recall. First, we introduce a notion of *trigger regret* in extensive-form games, which extends that of internal regret in normal-form games. When each player has low trigger regret, the empirical frequency of play is close to an EFCE. Then, we give an efficient no-trigger-regret algorithm. Our algorithm decomposes trigger regret into local subproblems at each decision point for the player, and constructs a global strategy of the player from the local solutions at each decision point.

## 1 Introduction

The *Nash equilibrium* (NE) [25] is the most common notion of rationality in game theory, and its computation in two-player, zero-sum games has been the flagship computational challenge in the area at the interplay between computer science and game theory (see, *e.g.*, the landmark results in heads-up no-limit poker by Brown and Sandholm [3] and Moravčík et al. [23]). The assumption underpinning NE is that the interaction among players is fully *decentralized*. Therefore, an NE is a distribution on the *uncorrelated* strategy space (*i.e.*, a product of independent distributions, one per player). A competing notion of rationality is the *correlated equilibrium* (CE) proposed by Aumann [1]. A *correlated strategy* is a general distribution over joint action profiles and it is customarily modeled via a trusted external *mediator* that draws an action profile from this distribution, and

---

[*]Equal contribution.

privately recommends to each player her component. A correlated strategy is a CE if no player has an incentive to choose an action different from the mediator's recommendation, because, assuming that all other players also obey, the suggested strategy is the best in expectation.

Many real-world strategic interactions involve more than two players with arbitrary (*i.e.*, general-sum) utilities. In these settings, the notion of NE presents some weaknesses which render the CE a natural solution concept: (i) computing an NE is an intractable problem, being PPAD-complete even in two-player games [7, 8]; (ii) the NE is prone to equilibrium selection issues; and (iii) the social welfare that can be attained via an NE may be significantly lower than what can be achieved via a CE [21, 28]. Moreover, in normal-form games, the notion of CE arises from simple learning dynamics in senses that NE does not [18, 6].

The notion of *extensive-form correlated equilibrium* (EFCE) by von Stengel and Forges [32] is a natural extension of the CE to the case of sequential strategic interactions. In an EFCE, the mediator draws, before the beginning of the sequential interaction, a recommended action for each of the possible decision points (*i.e.*, *information sets*) that players may encounter in the game, but she does not immediately reveal recommendations to each player. Instead, the mediator incrementally reveals relevant individual moves as players reach new information sets. At any decision point, the acting player is free to defect from the recommended action, but doing so comes at the cost of future recommendations, which are no longer issued if the player deviates.

**Original contributions**  We focus on general-sum extensive-form games with an arbitrary number of players (including the *chance player*). In this setting, the problem of computing a feasible EFCE can be solved in polynomial time in the size of the game tree [19] via a variation of the *Ellipsoid Against Hope* algorithm [26, 20]. However, in practice, this approach cannot scale beyond toy problems. Therefore, the following question remains open: *is it possible to devise simple dynamics leading to a feasible EFCE?* In this paper, we show that the answer is positive. To do so, we define an EFCE via the notion of *trigger agent* [17, 9]. Then, we define the notion of *trigger regret*, *i.e.*, a notion of *internal regret* suitable for extensive-form games. We provide an algorithm, which we call ICFR, that minimizes trigger agent regrets via the decomposition of these regrets locally at each information set. In order to do so, ICFR instantiates an internal regret minimizer and multiple external regret minimizers for each information set. We show that it is possible to orchestrate the learning procedure so that, for each information set, employing one regret minimizer per round does not compromise the overall convergence of the algorithm. The empirical frequency of play generated by ICFR converges to an EFCE almost surely in the limit. These results generalize the seminal work by Hart and Mas-Colell [18] to the sequential case via a simple and natural framework.

## 2 Preliminaries

In this section, we provide some groundings on sequential games and regret minimization (see the books by Shoham and Leyton-Brown [29] and Cesa-Bianchi and Lugosi [6], for additional details).

### 2.1 Extensive-form games

We focus on *extensive-form games* (EFGs) with imperfect information. We denote the set of players as $\mathcal{P} \cup \{c\}$, where $c$ is a *chance player* that selects actions according to fixed known probability distributions, representing exogenous stochasticity. An EFG is usually defined by means of a *game tree*, where $H$ is the set of nodes of the tree, and a node $h \in H$ is identified by the ordered sequence of actions from the root to the node. $Z \subseteq H$ is the set of terminal nodes, which are the leaves of the tree. For every $h \in H \setminus Z$, we let $P(h) \in \mathcal{P} \cup \{c\}$ be the unique player who acts at $h$ and $A(h)$ be the set of actions she has available. For each player $i \in \mathcal{P}$, we let $u_i : Z \to \mathbb{R}$ be her payoff function. Moreover, we denote by $p_c : Z \to (0, 1)$ the function assigning each terminal node $z \in Z$ to the product of probabilities of chance moves encountered on the path from the root of the game tree to $z$.

Imperfect information is encoded by using *information sets* (infosets). Given $i \in \mathcal{P}$, a player $i$'s infoset $I$ groups nodes belonging to player $i$ that are indistinguishable for her, *i.e.*, $A(h) = A(k)$ for any pair of nodes $h, k \in I$. $\mathcal{I}_i$ denotes the set of all player $i$'s infosets. Moreover, we let $A(I)$ be the set of actions available at infoset $I \in \mathcal{I}_i$. As customary, we assume that the game has *perfect recall*, *i.e.*, the infosets are such that no player forgets information once acquired. In EFGs with perfect recall, the infosets $\mathcal{I}_i$ of each player $i \in \mathcal{P}$ are partially ordered. We write $I \preceq J$

whenever infoset $I \in \mathcal{I}_i$ *precedes* $J \in \mathcal{I}_i$ according to such ordering, *i.e.*, formally, there exists a path in the game tree connecting a node $h \in I$ to some node $k \in J$. For the ease of notation, given $I \in \mathcal{I}_i$, we let $\mathcal{C}^\star(I)$ be the set of player $i$'s infosets that follow infoset $I$ (this included), defined as $\mathcal{C}^\star(I) := \{J \in \mathcal{I}_i \mid I \preceq J\}$. Moreover, given $I \in \mathcal{I}_i$ and $a \in A(I)$, we let $\mathcal{C}(I,a) \subseteq \mathcal{I}_i$ be the set of player $i$'s infosets that immediately follow $I$ by playing action $a$, *i.e.*, those reachable from at least one node $h \in I$ by following a path that includes $a$ and does not pass through another infoset of $i$.

**Normal-form plans and strategies**  A *normal-form plan* for player $i \in \mathcal{P}$ is a tuple $\pi_i \in \Pi_i := \bigtimes_{I \in \mathcal{I}_i} A(I)$ which specifies an action for each player $i$'s infoset, where $\pi_i(I)$ represents the action selected by $\pi_i$ at infoset $I \in \mathcal{I}_i$. We denote with $\pi \in \Pi := \bigtimes_{i \in \mathcal{P}} \Pi_i$ a *joint normal-form plan*, defining a plan $\pi_i \in \Pi_i$ for each player $i \in \mathcal{P}$. Moreover, a tuple defining normal-form plans for the opponents of player $i \in \mathcal{P}$ is denoted as $\pi_{-i} \in \Pi_{-i} := \bigtimes_{j \neq i \in \mathcal{P}} \Pi_j$. A *normal-form strategy* $\mu_i \in \Delta_{\Pi_i}$ is a probability distribution over $\Pi_i$, where $\mu_i[\pi_i]$ denotes the probability of selecting a plan $\pi_i \in \Pi_i$ according to $\mu_i$. Moreover, $\mu \in \Delta_\Pi$ is a *joint probability distribution* defined over $\Pi$, with $\mu[\pi]$ being the probability that the players end up playing the plans prescribed by $\pi \in \Pi$.

**Sequences**  For any player $i \in \mathcal{P}$, given an infoset $I \in \mathcal{I}_i$ and an action $a \in A(I)$, we denote with $\sigma = (I,a)$ the *sequence* of player $i$'s actions reaching infoset $I$ and terminating with $a$. Notice that, in EFGs with perfect recall, such sequence is uniquely determined, as paths that reach nodes belonging to the same infoset identify the same sequence of player $i$'s actions. We let $\Sigma_i := \{(I,a) \mid I \in \mathcal{I}_i, a \in A(I)\} \cup \{\varnothing_i\}$ be the set of player $i$'s sequences, where $\varnothing_i$ is the empty sequence of player $i$ (representing the case in which she never plays). Additionally, given an infoset $I \in \mathcal{I}_i$, we let $\sigma(I) \in \Sigma_i$ be the sequence of player $i$'s actions that identify infoset $I$.

**Subsets of (joint) normal-form plans**  We now define a few useful subsets of $\Pi_i$. The reader is encouraged to refer to Figure 1 for a simple example. For every player $i \in \mathcal{P}$ and infoset $I \in \mathcal{I}_i$, we let $\Pi_i(I) \subseteq \Pi_i$ be the set of player $i$'s normal-form plans that prescribe to play so as to reach infoset $I$ whenever possible (depending on the opponents' actions up to that point) and *any* action whenever reaching $I$ is *not* possible anymore. Moreover, for every sequence $\sigma = (I,a) \in \Sigma_i$, we let $\Pi_i(\sigma) \subseteq \Pi_i(I) \subseteq \Pi_i$ be the set of player $i$'s plans that reach infoset $I$ and recommend action $a$ at $I$. Similarly, given a terminal node $z \in Z$, we denote with $\Pi_i(z) \subseteq \Pi_i$ the set of normal-form plans by which player $i$ plays so as to reach $z$, while $\Pi(z) := \bigtimes_{i \in \mathcal{P}} \Pi_i(z)$ and $\Pi_{-i}(z) := \bigtimes_{j \neq i \in \mathcal{P}} \Pi_j(z)$.

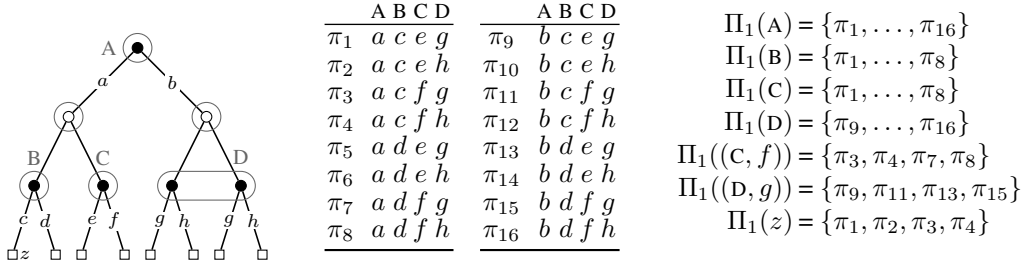

Figure 1: (Left) Sample game tree. Black round nodes belong to Player 1, white round nodes belong to Player 2, and white square nodes are leaves. Rounded, gray lines denote information sets. (Center) Set $\Pi_1$ of normal-form plans for Player 1. Each plan identifies an action at each information set. (Right) Examples of certain subsets of $\Pi_1$ defined in this subsection.

**Additional notation**  For every $i \in \mathcal{P}$ and $I \in \mathcal{I}_i$, we let $Z(I) \subseteq Z$ be the set of terminal nodes that are reachable from infoset $I \in \mathcal{I}_i$ of player $i$. Moreover, $Z(I,a) \subseteq Z(I) \subseteq Z$ is the set of terminal nodes reachable by playing action $a \in A(I)$ at infoset $I$, whereas $Z^c(I,a) := Z(I) \setminus Z(I,a)$ is the set of terminal nodes which are reachable by playing an action different from $a$ at $I$. For any player $i \in \mathcal{P}$, normal-form plan $\pi_i \in \Pi_i$, infoset $I \in \mathcal{I}_i$, and terminal node $z \in Z$, we define $\rho_{I \to z}^{\pi_i}$ as a function equal to 1 if $z$ is reachable from $I$ when player $i$ plays according to $\pi_i$, and 0 otherwise. Finally, we define a notion of *reach* such that, for each normal-form plan $\pi = (\pi_i, \pi_{-i}) \in \Pi$, infoset $I \in \mathcal{I}_i$, and terminal node $z \in Z$, we have $\rho_{I \to z}^{(\pi_i, \pi_{-i})} := \rho_{I \to z}^{\pi_i} \cdot \mathbb{1}[\pi_{-i} \in \Pi_{-i}(z)]$.

## 2.2 External and internal regret minimization

In the regret minimization framework [33], each player $i \in \mathcal{P}$ plays repeatedly against the others by making a series of decisions from a set $\mathcal{X}_i$. A *regret minimizer* for player $i \in \mathcal{P}$ is a device that, at each iteration $t = 1, \ldots, T$, supports two operations: (i) RECOMMEND, which provides the next decision $x_i^{t+1} \in \mathcal{X}_i$ on the basis of the past history of play and the observed utilities up to iteration $t$; and (ii) OBSERVE, which receives a utility function $u_i^t : \mathcal{X}_i \to \mathbb{R}$ that is used to evaluate decision $x_i^t$. A regret minimizer is evaluated in terms of its cumulative regret. Two types of regret minimizers are commonly studied, depending on the adopted notion of regret, either *external* or *internal* regret.

**External regret**  An *external-regret minimizer* $\mathcal{R}^{\text{EXT}}$ for player $i \in \mathcal{P}$ is a device minimizing the *cumulative external regret* of player $i$ up to iteration $T$, which is defined as:

$$R_i^T := \max_{\hat{x}_i \in \mathcal{X}_i} \left\{ \sum_{t=1}^T u_i^t(\hat{x}_i) \right\} - \sum_{t=1}^T u_i^t(x_i^t). \tag{1}$$

$R_i^T$ represents how much player $i$ would have gained by always taking the best decision in hindsight, given the history of utilities observed up to iteration $T$.

**Internal regret**  An *internal-regret minimizer* $\mathcal{R}^{\text{INT}}$ for player $i \in \mathcal{P}$ is a device minimizing the *cumulative internal regret* of player $i$ up to iteration $T$, which is defined as:

$$\max_{x_i, \hat{x}_i \in \mathcal{X}_i} R_{i,(x_i,\hat{x}_i)}^T := \max_{x_i, \hat{x}_i \in \mathcal{X}_i} \left\{ \sum_{t=1}^T \mathbb{1}[x_i = x_i^t] \big( u_i^t(\hat{x}_i) - u_i^t(x_i) \big) \right\}. \tag{2}$$

Intuitively, player $i$ has small internal regret if, for each pair of decisions $(x_i, \hat{x}_i)$, she does not regret of not having played $\hat{x}_i$ each time she selected $x_i$. The notion of internal regret is strictly stronger than the notion of external regret: any algorithm with small internal regret also has small external regret, but the converse does not hold (see Stoltz and Lugosi [31] for an example).

Regret minimizers show an interesting connection with games when the decision sets $\mathcal{X}_i$ are the sets of normal-form plans $\Pi_i$ and the observed utilities $u_i^t$ are obtained by playing the game according to the selected plans $\pi_i^t$. Letting $\pi^t := (\pi_i^t)_{i \in \mathcal{P}}$ be the joint normal-form plan resulting at each iteration $t = 1, \ldots, T$, we denote with $\{\pi^t\}_{t=1}^T$ the overall sequence of plays made by the players. Then, the *empirical frequency of play* $\bar{\mu}^T \in \Delta_\Pi$ generated by $\{\pi^t\}_{t=1}^T$ is such that for every $\pi \in \Pi$:

$$\bar{\mu}^T(\pi) := \frac{|\{1 \leq t \leq T \mid \pi^t = \pi\}|}{T}. \tag{3}$$

If all the players play according to some external-regret minimizers, then $\bar{\mu}^T$ approaches the set of (normal-form) coarse correlated equilibria, even in EFGs (see Cesa-Bianchi and Lugosi [6] and Celli et al. [5] for further details). Moreover, Foster and Vohra [16] and Hart and Mas-Colell [18] established that the empirical frequency of play generated by any no-internal-regret algorithm (see Cesa-Bianchi and Lugosi [6] and Blum and Mansour [2] for some examples) converges to the set of correlated equilibria in repeated games with simultaneous moves (*i.e.,* normal-form games).

## 3 Extensive-form correlated equilibria

The definition of EFCE requires the following notion of trigger agent, which, intuitively, is associated to each player and each of her sequences of action recommendations.

**Definition 1** (Trigger agent for EFCE). *Given a player $i \in \mathcal{P}$, a sequence $\sigma = (I, a) \in \Sigma_i$, and a probability distribution $\hat{\mu}_i \in \Delta_{\Pi_i(I)}$, an $(\sigma, \hat{\mu}_i)$-trigger agent for player $i$ is an agent that takes on the role of player $i$ and commits to following all recommendations unless she reaches $I$ and gets recommended to play $a$. If this happens, the player stops committing to the recommendations and plays according to a plan sampled from $\hat{\mu}_i$ until the game ends.*

It follows that joint probability distribution $\mu \in \Delta_\Pi$ is an EFCE if, for every $i \in \mathcal{P}$, player $i$'s expected utility when following the recommendations is at least as large as the expected utility that any $(\sigma, \hat{\mu}_i)$-trigger agent for player $i$ can achieve (assuming the opponents' do not deviate).

For any $\mu \in \Delta_\Pi$, sequence $\sigma = (I, a) \in \Sigma_i$, and $(\sigma, \hat{\mu}_i)$-trigger agent, we define the probability of the game ending in a terminal node $z \in Z(I)$ as:

$$p_{\mu,\hat{\mu}_i}^\sigma(z) := \left( \sum_{\substack{\pi_i \in \Pi_i(\sigma) \\ \pi_{-i} \in \Pi_{-i}(z)}} \mu(\pi_i, \pi_{-i}) \right) \left( \sum_{\hat{\pi}_i \in \Pi_i(z)} \hat{\mu}_i(\hat{\pi}_i) \right) p_c(z), \qquad (4)$$

which accounts for the fact that the agent follows recommendations until she receives the recommendation of playing $a$ at $I$, and, thus, she 'gets triggered' and plays according to $\hat{\pi}_i$ sampled from $\hat{\mu}_i$ from $I$ onwards. Moreover, the probability of reaching a terminal node $z \in Z(I, a)$ when following the recommendations is defined as follows:

$$q_\mu(z) := \left( \sum_{\pi \in \Pi(z)} \mu(\pi) \right) p_c(z). \qquad (5)$$

The definition of EFCE reads as follows (see Appendix A or the work by Farina et al. [13] for details):

**Definition 2** (Extensive-form correlated equilibrium). *An EFCE of an EFG is a joint probability distribution $\mu \in \Delta_\Pi$ such that, for every $i \in \mathcal{P}$ and $(\sigma, \hat{\mu}_i)$-trigger agent for player $i$, with $\sigma = (I, a) \in \Sigma_i$, it holds:*

$$\sum_{z \in Z(I,a)} q_\mu(z) u_i(z) \geq \sum_{z \in Z(I)} p_{\mu,\hat{\mu}_i}^\sigma(z) u_i(z). \qquad (6)$$

A joint probability distribution $\mu \in \Delta_\Pi$ is said to be an $\epsilon$-EFCE when the *maximum deviation* $\delta(\mu)$ under $\mu$ is such that:

$$\delta(\mu) := \max_{i \in \mathcal{P}} \max_{\sigma = (I,a) \in \Sigma_i} \left\{ \max_{\hat{\mu}_i \in \Delta_{\Pi_i(I)}} \left\{ \sum_{z \in Z(I)} p_{\mu,\hat{\mu}_i}^\sigma(z) u_i(z) \right\} - \sum_{z \in Z(I,a)} q_\mu(z) u_i(z) \right\} \leq \epsilon. \qquad (7)$$

## 4 Trigger regret and relationships with EFCE

In this section, we introduce the notion of *trigger regret*. Intuitively, it measures the regret that each trigger agent has for not having played the best-in-hindsight strategy. As we will show, when each trigger agent has low trigger regret, then the empirical frequency of play is close to being an EFCE.

Given a sequence $\{\pi^t\}_{t=1}^T$, the vector of *immediate utilities* $u_i^t$ observed by player $i \in \mathcal{P}$ after any iteration $t = 1, \ldots, T$ is defined as follows. For every infoset $I \in \mathcal{I}_i$ and action $a \in A(I)$ we have:

$$u_i^t[I, a] := \sum_{z \in Z(I,a) \setminus \bigcup_{J \in \mathcal{C}(I,a)} Z(J)} \mathbb{1}[\pi_{-i}^t \in \Pi_{-i}(z)] \, p_c(z) u_i(z),$$

which represents the utility experienced by player $i$ if the game ends after playing action $a$ at infoset $I$, without going through other player $i$'s infosets and assuming that the other players play as prescribed by the plans $\pi_{-i}^t \in \Pi_{-i}$ at iteration $t$. Notice that the summation is over the terminal nodes immediately reachable from $I$ by playing $a$ and the payoff of each terminal node is multiplied by the probability of reaching it given chance probabilities.

For $i \in \mathcal{P}$, the following recursive formula defines player $i$'s utility attainable at infoset $I \in \mathcal{I}_i$ when a normal-form plan $\pi_i \in \Pi_i$ is selected:

$$V_I^t(\pi_i) := u_i^t[I, \pi_i(I)] + \sum_{J \in \mathcal{C}(I, \pi_i(I))} V_J^t(\pi_i). \qquad (8)$$

**Definition 3** (Trigger regret). *For every player $i \in \mathcal{P}$ and sequence $\sigma = (I, a) \in \Sigma_i$, we let $R_\sigma^T$ be the* trigger regret *for sequence $\sigma$, which we define as follows:*

$$R_\sigma^T := \max_{\hat{\pi}_i \in \Pi_i(I)} \left\{ \sum_{t=1}^T \mathbb{1}[\pi_i^t \in \Pi_i(\sigma)] \left( V_I^t(\hat{\pi}_i) - V_I^t(\pi_i^t) \right) \right\}.$$

The trigger regret for $\sigma = (I, a)$ represents the regret experienced by the trigger agent that gets triggered on sequence $\sigma$, *i.e.*, when infoset $I$ is reached and action $a$ is recommended. Notice that $R_\sigma^T$ only accounts for those iterations in which $\pi_i^t \in \Pi_i(\sigma)$, *i.e.*, intuitively, when the actions prescribed by the normal-form plan $\pi_i^t$ trigger the agent associated to sequence $\sigma$.

The following theorem shows that minimizing the trigger regrets for each player $i \in \mathcal{P}$ and sequence $\sigma \in \Sigma_i$ allows to approach the set of EFCEs.

**Theorem 1.** *At all times $T$, the empirical frequency of play $\bar{\mu}^T$ (Equation 3) is an $\epsilon$-EFCE, where*

$$\epsilon := \max_{i \in \mathcal{P}} \max_{\sigma \in \Sigma_i} \frac{R_\sigma^T}{T}.$$

**Corollary 1.** *If $\limsup_{T \to \infty} \max_{i \in \mathcal{P}} \max_{\sigma \in \Sigma_i} \frac{R_\sigma^T}{T} \leq 0$, then $\limsup_{T \to \infty} \delta(\bar{\mu}^T) \leq 0$, that is, for any $\epsilon > 0$, eventually the empirical frequency of play $\bar{\mu}^T$ becomes an $\epsilon$-EFCE.*

## 5 Laminar regret decomposition for trigger regret

In order to design an algorithm minimizing trigger regrets, we first develop a new regret decomposition that extends the *laminar regret decomposition* framework introduced by Farina et al. [12]. Our decomposition exploits the structure of the EFG to show that trigger regrets can be minimized by minimizing other suitably defined regret terms which are *local* at each infoset.

First, for each player $i \in \mathcal{P}$, sequence $\sigma = (J, a) \in \Sigma_i$, and infoset $I \in \mathcal{C}^\star(J)$ (*i.e.*, any infoset following from $J$, this included), we define the notion of *subtree regret* as follows:

$$R_{\sigma,I}^T := \max_{\hat{\pi}_i \in \Pi_i(I)} \left\{ \sum_{t=1}^T \mathbb{1}[\pi_i^t \in \Pi_i(\sigma)] \left( V_I^t(\hat{\pi}_i) - V_I^t(\pi_i^t) \right) \right\}.$$

Each term $R_{\sigma,I}^T$ represents the regret at infoset $I$ experienced by the trigger agent that gets triggered on sequence $\sigma = (J, a)$. Differently from the trigger regret $R_\sigma^T$, which is defined only for the infoset $J$ of $\sigma$, the subtree regrets $R_{\sigma,I}^T$ are defined for all the infosets $I \in \mathcal{I}_i$ such that $J \preceq I$.

**Remark 1.** *Given player $i \in \mathcal{P}$, it is immediate to see that, if $R_{\sigma,I}^T = o(T)$ for each $\sigma = (J, a) \in \Sigma_i$ and $I \in \mathcal{C}^\star(J)$, then $R_\sigma^T = o(T)$ for every $\sigma \in \Sigma_i$. Therefore, we can safely focus on the problem of minimizing subtree regrets, as this will automatically guarantee convergence to an EFCE.*

Next, we need to introduce, for every player $i \in \mathcal{P}$ and infoset $I \in \mathcal{I}_i$, the following parameterized utility function defined at each iteration $t = 1, \ldots, T$:

$$\hat{u}_I^t : A(I) \ni a \mapsto u_i^t[I, a] + \sum_{J \in \mathcal{C}(I,a)} V_J^t(\pi_i^t), \tag{9}$$

which represents the utility that player $i$ gets, at iteration $t$, by playing action $a$ at $I$ and following the actions prescribed by $\pi_i^t$ at the subsequent infosets. Then, for each sequence $\sigma = (J, a') \in \Sigma_i$, infoset $I \in \mathcal{C}^\star(J)$, and action $a \in A(I)$, the *laminar subtree regret of action $a$* is defined as:

$$\hat{R}_{\sigma,I,a}^T := \sum_{t=1}^T \mathbb{1}[\pi_i^t \in \Pi_i(\sigma)] \left( \hat{u}_I^t(a) - \hat{u}_I^t(\pi_i^t(I)) \right), \tag{10}$$

while, for $\sigma = (J, a') \in \Sigma_i$ and $I \in \mathcal{C}^\star(J)$, the *laminar subtree regret* is:

$$\hat{R}_{\sigma,I}^T := \max_{a \in A(I)} \hat{R}_{\sigma,I,a}^T. \tag{11}$$

The following two lemmas show that the subtree regrets can be minimized by minimizing the laminar subtree regrets at all the infosets of the game.

**Lemma 1.** *The subtree regret for each player $i \in \mathcal{P}$, sequence $\sigma = (J, a') \in \Sigma_i$, and infoset $I \in \mathcal{C}^\star(J)$ can be decomposed as:*

$$R_{\sigma,I}^T = \max_{a \in A(I)} \left\{ \hat{R}_{\sigma,I,a}^T + \sum_{I' \in \mathcal{C}(I,a)} R_{\sigma,I'}^T \right\}.$$

The lemma is proved by recursively applying the definitions of $R_{\sigma,I}^T$ and $V_I^t(\hat{\pi}_i)$, and by exploiting Equation (9). Then, Lemma 1 is used to show the following.

**Lemma 2.** *For every player $i \in \mathcal{P}$, sequence $\sigma = (J, a') \in \Sigma_i$, and infoset $I \in \mathcal{C}^\star(J)$, it holds:*

$$R_{\sigma,I}^T \le \max_{\hat{\pi}_i \in \Pi_i(I)} \sum_{I' \in \mathcal{C}^\star(I)} \mathbb{1}[\hat{\pi}_i \in \Pi_i(I')] \, \hat{R}_{\sigma,I'}^T. \tag{12}$$

# 6 Internal counterfactual regret minimization

We propose the *internal counterfactual regret minimization* algorithm (ICFR) as a way to minimize the laminar subtree regrets described in the previous section. At each iteration $t$, ICFR builds a normal-form plan $\pi_i^t$ in a top-down fashion by sampling an action locally at each infoset, following a simple rule: if the current infoset can be reached through $\pi_i^t$, then an action is sampled according to an internal-regret minimizer; otherwise, an external-regret minimizer is employed.

In order to minimize the laminar subtree regrets, ICFR needs to instantiate different regret minimizers for each infoset. For every infoset $I \in \mathcal{I}_i$, the algorithm instantiates an internal-regret minimizer $\mathcal{R}_I^{\text{INT}}$ employing an arbitrary no-internal-regret algorithm. Moreover, let $\Sigma_i^c(I) \subseteq \Sigma_i$ be the set of sequences of player $i$ that do not allow to reach $I$ and whose last action is played at an infoset preceding $I$. Formally,

$$\Sigma_i^c(I) := \{(J, a) \in \Sigma_i \mid J \preceq I, a \notin \sigma(I)\}.$$

ICFR instantiates an additional external-regret minimizer $\mathcal{R}_{\sigma,I}^{\text{EXT}}$ for each sequence $\sigma \in \Sigma_i^c(I)$. The internal-regret minimizer $\mathcal{R}_I^{\text{INT}}$ is responsible for the minimization of the laminar subtree regrets $\hat{R}_{\sigma,I}^T$ associated to trigger sequences $\sigma = (I, a) \in \Sigma_i$ for each $a \in A(I)$. Instead, the external-regret minimizers $\mathcal{R}_{\sigma,I}^{\text{EXT}}$ are responsible for the laminar subtree regrets of sequences $\sigma \in \Sigma_i^c(I)$.

Algorithm 1 provides a description of the procedures adopted by ICFR. At iteration $t$ and for each

---

**Algorithm 1** ICFR (for Player $i$)

1: **function** ICFR($i$)
2:     Initialize the regret minimizers
3:     $t \leftarrow 1$
4:     **while** $t < T$ **do**
5:         $\pi_i^t \leftarrow$ SAMPLEINTERNAL
6:         Observe $u_i^t$ (*i.e.*, $u_i^t[I, a]$ for each pair $(I, a)$)
7:         UPDATEINTERNAL($\pi_i^t, u_i^t$)
8:         $t \leftarrow t + 1$
9: **function** SAMPLEINTERNAL
10:     **for** $I \in \mathcal{I}_i$ in a top-down order **do**
11:         **if** $\pi_i^t \in \Pi_i(I)$ **then**
12:             $\pi_i^t(I) \leftarrow \mathcal{R}_I^{\text{INT}}$.RECOMMEND()
13:         **else**
14:             $\sigma_I^t \leftarrow \Sigma_i^c(I) \cap \{(J, \pi_i^t(J)) \mid J \preceq I\}$
15:             $\pi_i^t(I) \leftarrow \mathcal{R}_{\sigma_I^t, I}^{\text{EXT}}$.RECOMMEND()
16: **function** UPDATEINTERNAL($\pi_i^t, u_i^t$)
17:     **for** $I \in \mathcal{I}_i$ **do**
18:         $\mathcal{R}_I^{\text{INT}}$.OBSERVE($\mathbb{1}[\pi_i^t \in \Pi_i(I)] \cdot \hat{u}_I^t$)
19:         **for** $\sigma \in \Sigma_i^c(I)$ **do**
20:             $\mathcal{R}_{\sigma,I}^{\text{EXT}}$.OBSERVE($\mathbb{1}[\pi_i^t \in \Pi_i(\sigma)] \cdot \hat{u}_I^t$)

---

$I \in \mathcal{I}_i$, an action is sampled as follows: if the (possibly partial) normal-form plan $\pi_i^t$ sampled up to this point allows $I$ to be reached (*i.e.*, it is still possible that $\pi_i^t \in \Pi_i(I)$), then an action is selected according to the internal-regret minimizer $\mathcal{R}_I^{\text{INT}}$ (Line 12). Otherwise, if $I$ cannot be reached through the (possibly partial) plan $\pi_i^t$, then we let $\sigma_I^t$ be the unique sequence in $\Sigma_i^c(I)$ whose actions are prescribed by $\pi_i^t$ (Line 14). In this case, the player follows the strategy recommended by the external-regret minimizer $\mathcal{R}_{\sigma_I^t, I}^{\text{EXT}}$ (Line 15). In the update procedure, the regret minimizers are fed with the vectors $\hat{u}_I^t$, which, with an abuse of notation, denote the vectors whose components are defined by the values of the corresponding parameterized utility functions $\hat{u}_I^t$ in Equation (9). In particular, for each $I \in \mathcal{I}_i$, the internal-regret minimizer $\mathcal{R}_I^{\text{INT}}$ observes the utility vector $\hat{u}_I^t$ only if the sampled plan $\pi_i^t$ allows to reach infoset $I$, while each external-regret minimizer $\mathcal{R}_{\sigma,I}^{\text{EXT}}$ is updated only if $\pi_i^t$ prescribes all the actions in the corresponding sequence $\sigma$ (Line 18 and Line 20, respectively).

The crucial insight is that for each infoset $I \in \mathcal{I}_i$, no matter the action selected at $I$, only one of the regret minimizers will receive a non-zero utility. Consequently, only one of the regret minimizers can cumulate regret at time $t$, and that is the regret whose recommendation we follow. Therefore, it is possible to show that the empirical frequency of play $\bar{\mu}^T$ obtained via ICFR converges almost surely to an EFCE. We start with the following auxiliary result.

**Lemma 3.** *For any $I, J \in \mathcal{I}_i : I \preceq J$, if $\hat{R}_{\sigma,J}^T = o(T)$ for all $\sigma = (I, a) \in \Sigma_i$ then $\hat{R}_{\sigma(I),J}^T = o(T)$.*

Then, our main result reads as follows:

**Theorem 2.** *When all the players play according to* ICFR, *$\bar{\mu}^T$ converges almost surely to an EFCE.*

**Example** We provide a simple example illustrating the key ideas of ICFR. Figure 2–Left describes an EFG with two infosets $I, J$ of the same player (player $i$). Even in such a simple setting ICFR has to ensure that six laminar subtree regrets are properly minimized (see Figure 2–Right). To simplify the notation, throughout the example we write $\hat{R}^T_{a,I}$ in place of $\hat{R}^T_{(I,a),J}$ (the remaining regrets are treated analogously). ICFR instantiates one internal-regret minimizer for each infoset of player $i$. We denote them by $\mathcal{R}^{\text{INT}}_I$ and $\mathcal{R}^{\text{INT}}_J$, respec-

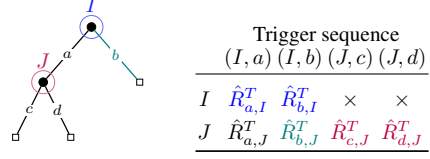

Trigger sequence

| | $(I,a)$ | $(I,b)$ | $(J,c)$ | $(J,d)$ |
|---|---|---|---|---|
| $I$ | $\hat{R}^T_{a,I}$ | $\hat{R}^T_{b,I}$ | $\times$ | $\times$ |
| $J$ | $\hat{R}^T_{a,J}$ | $\hat{R}^T_{b,J}$ | $\hat{R}^T_{c,J}$ | $\hat{R}^T_{d,J}$ |

Figure 2: (Left) EFG with two infosets $I$ and $J$ of player $i$. (Right) The laminar subtree regrets.

tively. Then, we observe that $\Sigma^c_i(J) = \{(I,b)\}$, because $b$ is the only action of player $i$ satisfying the following conditions: (i) it departs from an infoset which is on the path from the root node to $J$ and (ii) if player $i$ selected $b$ at infoset $I$, she would no longer be able to reach $J$. Therefore, ICFR instantiates the external-regret minimizer $\mathcal{R}^{\text{EXT}}_{b,J}$.

Suppose to be at iteration $t$ of ICFR. The sampling procedure starts from infoset $I$. Being the root of the EFG, $I$ is always reached by player $i$. Therefore, an action is selected following the recommendation of the internal-regret minimizer $\mathcal{R}^{\text{INT}}_I$. During the update procedure, $\mathcal{R}^{\text{INT}}_I$ is provided with the utility resulting from the normal-form plan $\pi^t_i$ obtained from the sampling procedure. Intuitively, this ensures that $\hat{R}^T_{a,I}$ and $\hat{R}^T_{b,I}$ are small. Now, there are two possibilities:

**Case $\pi^t_i(I) = a$.** The partial plan $\pi^t_i$ allows $J$ to be reached. Therefore, at $J$, an action is chosen according to the strategy recommended by $\mathcal{R}^{\text{INT}}_J$. Then, in the update procedure, the internal-regret minimizer $\mathcal{R}^{\text{INT}}_J$ is provided with the observed utility, while the external-regret minimizer is not updated. This ensures that $\hat{R}^T_{c,J}$ and $\hat{R}^T_{d,J}$ are managed properly. By Equation 11, the choice at $t$ does not impact $\hat{R}^T_{b,J}$ since $\pi^t_i \notin \Pi_i(I,b)$, while $\hat{R}^T_{a,J}$ is affected by the choice at $J$ because $a \in \sigma(J)$. The internal-regret minimizer $\mathcal{R}^{\text{INT}}_J$ guarantees that $\hat{R}^T_{c,J} = o(T)$ and $\hat{R}^T_{d,J} = o(T)$. Then, by using Lemma 3, we have that $\hat{R}^T_{a,J} = o(T)$ holds as well.

**Case $\pi^t_i(I) = b$.** We have that $\sigma^t_J = (I,b)$. An action at $J$ is sampled according to the external-regret minimizer $\mathcal{R}^{\text{EXT}}_{b,J}$, which is then provided with the observed utility (the internal-regret minimizer $\mathcal{R}^{\text{INT}}_J$ is not updated). This ensures that the increase in $\hat{R}^T_{b,J}$ is small. The other regret terms are not impacted by the choice at $t$.

## 7 Experimental evaluation

We evaluate the convergence of ICFR on the standard benchmark games for the computation of correlated equilibria. We use parametric instances from four different multi-player games: Kuhn poker [22], Leduc poker [30], Goofspiel [27], and Battleship [13]. Instances of the Kuhn, Leduc, and Goofspiel games are parametric in the number of players $p$ and in the number of card ranks $r$. To increase the readability, we denote by K$p.r$ the Kuhn poker instance with $p$ players and $r$ ranks (the other instances are treated analogously). Our Battleship instance (denoted by B$\text{s}$) has a grid of size $2 \times 2$ and maximum number of rounds per player equal to 3. A detailed description of the games is provided in Appendix C.1. We use *Regret matching* [18] for external-regret minimizers, and the no-internal-regret algorithm by Blum and Mansour [2] for internal-regret minimizers. All experiments are run on a 64-core machine with 512 GB of RAM.

**Convergence of ICFR** Figure 3–Center displays the maximum deviation $\delta(\bar{\mu}^T)$ as a function of the number of rounds $T$. According to Equation (7), the strategy $\bar{\mu}^T$ is guaranteed to be a $\delta(\bar{\mu}^T)$-EFCE. We set a maximum number of $10^4$ iterations and, for each instance, we provide the average and the standard deviation computed over 50 different seeds. First, we notice that ICFR attains roughly an empirical convergence rate of $O(1/T)$. The performance over the Battleship instance suggests that equilibria with large support size are significantly more challenging to be computed. Second, we remark that, unlike recent algorithms for computing EFCEs by Farina et al. [13, 14], ICFR can be applied to games with more than two players including chance. Moreover, since EFCE $\subseteq$ EFCCE $\subseteq$ NFCCE, ICFR also provides a flexible way to compute $\epsilon$-EFCCEs and $\epsilon$-NFCCEs. In the former case, the only known algorithm can only handle games with two players

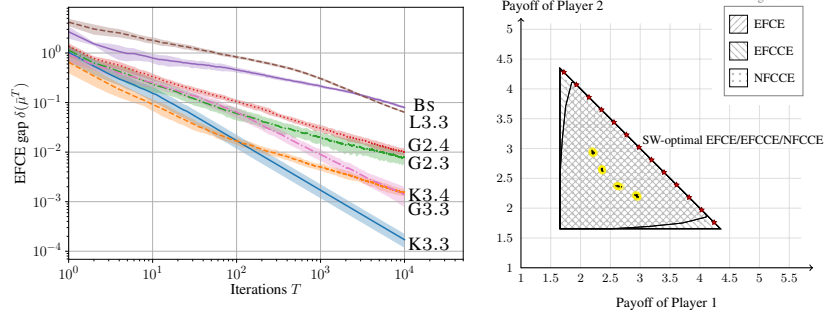

Figure 3: (Left) Dimension of the game instances in terms of number of players and infosets/sequences for each player. (Center) Convergence of ICFR. (Right) Social welfare attained at different $\epsilon$-EFCEs computed via ICFR (black dots corresponds to different seeds).

and no chance [15]. In the latter case, the recent algorithms by Celli et al. [5] are significantly outperformed. For example, previous algorithms cannot reach a 0.1-NFCCE in less than 24h on a Leduc instance with 1200 total infosets and a one-bet maximum per bidding round. ICFR reaches $\epsilon = 0.1$ in around 9h on an arguably more complex Leduc instance (*i.e.*, more than 9k total infosets and a two-bet maximum per round). Further details on the computation of EFCCEs and NFCCEs are provided in Appendix C.2, together with the plots of the decoupled EFCE deviations of each player.

**Social Welfare**  Figure 3–Right provides a visual depiction of the *quality* of the solutions attained by ICFR in terms of their social welfare. The figure displays the payoffs obtained for 100 different seeds in a two-player Goofspiel instance without chance (*i.e.,* the prize deck is sorted).

# Broader Impact

Correlated equilibria provide an appropriate solution concept for coordination problems in which agents have arbitrary utilities, and may work towards different objectives. The study of uncoupled dynamics converging to correlated equilibria in problems with sequential actions and hidden information lays new theoretical foundations for multi-agent reinforcement learning problems. Most of the work in the multi-agent reinforcement learning community either studies fully competitive settings, where agents play selfishly to reach a Nash equilibrium, or fully cooperative scenarios in which agents have the exact same goals. Our work could enable techniques that are in-between these two extremes: agents have arbitrary objectives, but coordinate their actions towards an equilibrium with some desired properties.

As we argued in the paper, the social welfare that can be attained via a Nash equilibrium (that is, by playing selfishly) may be significantly lower than what can be achieved via a correlated equilibrium. We provided some empirical evidences that ICFR computes equilibria which attain a social welfare 'not too far' from the optimal one. This could have an arguably positive societal impact when applied to real economic problems. However, further research in this direction is required to prevent 'winner-takes-all' scenarios in problems with an unbalanced reward structure where equilibria with high social welfare may just award players with the largest utilities at the expense of the others. This could provide a way to reach *fair* equilibria both in theory and in practice.

# Acknowledgments and Disclosure of Funding

This work is based on work supported by the Italian MIUR PRIN 2017 Project ALGADIMAR "Algorithms, Games, and Digital Market", the National Science Foundation under grants IIS-1718457, IIS-1617590, IIS-1901403, and CCF-1733556, and the ARO under awards W911NF-17-1-0082 and W911NF2010081. Gabriele Farina is supported by a Facebook fellowship.

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
