[Supplementary Material]

# A Extensive-form correlated equilibrium

In the context of EFGs, the two most widely adopted notions of correlated equilibrium are the *normal-form correlated equilibrium* (NFCE) [1] and the *extensive-form correlated equilibrium* (EFCE) [32]. In the former, the mediator draws and recommends a complete normal-form plan to each player before the game starts. Then, each player decides whether to follow the recommended plan or deviate to an arbitrary strategy she desires. In an EFCE, the mediator draws a normal-form plan for each player before the beginning of the game, but she does not immediately reveal it to each player. Instead, the mediator incrementally reveals individual moves as players reach new infosets. At any infoset, the acting player is free to deviate from the recommended action, but doing so comes at the cost of future recommendations, which are no longer issued if the player deviates.

In an EFCE, players know less about the normal-form plans that were sampled by the mediator than in an NFCE, where the whole normal-form plan is immediately revealed. Therefore, by exploiting an EFCE, the mediator can more easily incentivize players to follow strategies that may hurt them, as long as players are indifferent as to whether or not to follow the recommendations. This is beneficial when the mediator wants to maximize, *e.g.*, the social-welfare of the game.

A *coarse* correlated equilibrium enforces protection against deviations which are independent of the recommended move. *Normal-form coarse correlated equilibria* (NFCCEs) [24, 4] and *extensive-form coarse correlated equilibria* (EFCCEs) [15] are the coarse equivalent of NFCE and EFCE, respectively. For arbitrary EFGs with perfect recall, the following inclusion of the set of equilibria holds: NFCE $\subseteq$ EFCE $\subseteq$ EFCCE $\subseteq$ NFCCE [32, 15].

Appendix A.1 provides a suitable formal definition of the set of EFCEs via the notion of *trigger agent* (originally introduced by Gordon et al. [17] and Dudík and Gordon [9]). Finally, Appendix A.2 summarizes existing approaches for computing EFCEs.

## A.1 Formal definition of the set of EFCEs

The definition requires the following notion of trigger agent, which, intuitively, is associated to each player and each of her sequences of action recommendations.

**Definition 4** (Trigger agent for EFCE). *Given a player $i \in \mathcal{P}$, a sequence $\sigma = (I, a) \in \Sigma_i$ , and a probability distribution $\hat{\mu}_i \in \Delta_{\Pi_i(I)}$, an $(\sigma, \hat{\mu}_i)$-trigger agent for player $i$ is an agent that takes on the role of player $i$ and commits to following all recommendations unless she reaches $I$ and gets recommended to play $a$. If this happens, the player stops committing to the recommendations and plays according to a plan sampled from $\hat{\mu}_i$ until the game ends.*

It follows that joint probability distribution $\mu \in \Delta_\Pi$ is an EFCE if, for every $i \in \mathcal{P}$, player $i$'s expected utility when following the recommendations is at least as large as the expected utility that any $(\sigma, \hat{\mu}_i)$-trigger agent for player $i$ can achieve (assuming the opponents' do not deviate).

Given $\sigma = (I, a) \in \Sigma_i$, in order to express the expected utility of a $(\sigma, \hat{\mu}_i)$-trigger agent, it is convenient to define the probability of the game ending in each terminal node $z \in Z$. Three cases are possible. In the first one, $z \in Z(I, a)$. The probability of reaching $z$ given the joint probability distribution $\mu \in \Delta_\Pi$ and a $(\sigma, \hat{\mu}_i)$-trigger agent is defined as:

$$p^\sigma_{\mu, \hat{\mu}_i}(z) := \left( \sum_{\substack{\pi_i \in \Pi_i(\sigma) \\ \pi_{-i} \in \Pi_{-i}(z)}} \mu(\pi_i, \pi_{-i}) \right) \left( \sum_{\hat{\pi}_i \in \Pi_i(z)} \hat{\mu}_i(\hat{\pi}_i) \right) p_c(z), \tag{13}$$

which accounts for the fact that the agent follows recommendations until she receives the recommendation of playing $a$ at $I$, and, thus, she 'gets triggered' and plays according to $\hat{\pi}_i$ sampled from $\hat{\mu}_i$ from $I$ onwards. The second case is $z \in Z^c(I, a)$, which is reached with probability:

$$y^\sigma_{\mu, \hat{\mu}_i}(z) := \left( \sum_{\substack{\pi_i \in \Pi_i(\sigma) \\ \pi_{-i} \in \Pi_{-i}(z)}} \mu(\pi_i, \pi_{-i}) \right) \left( \sum_{\hat{\pi}_i \in \Pi_i(z)} \hat{\mu}_i(\hat{\pi}_i) \right) p_c(z) + \left( \sum_{\pi \in \Pi(z)} \mu(\pi) \right) p_c(z), \tag{14}$$

where the first term accounts for the event that $z$ is reached when the agent 'gets triggered', while the second term is the probability of reaching $z$ while not being triggered (notice that the two events are independent). Finally, the

third case is when $z \in Z \setminus Z(I)$ and the infoset $I$ is never reached. Then, the probability of reaching $z$ is defined as:

$$q_\mu(z) := \left( \sum_{\pi \in \Pi(z)} \mu(\pi) \right) p_c(z). \tag{15}$$

By exploiting the above definitions, the definition of EFCE reads as follows.

**Definition 5** (Extensive-form correlated equilibrium). *An EFCE of an EFG is a probability distribution $\mu \in \Delta_\Pi$ such that, for every $i \in \mathcal{P}$ and $(\sigma, \hat{\mu}_i)$-trigger agent for player $i$, with $\sigma = (I, a) \in \Sigma_i$, it holds:*

$$\sum_{z \in Z} \left( \sum_{\pi \in \Pi(z)} \mu(\pi) \right) p_c(z) u_i(z) \geq \sum_{z \in Z(I,a)} p^\sigma_{\mu,\hat{\mu}_i}(z) u_i(z) + \sum_{z \in Z^c(I,a)} y^\sigma_{\mu,\hat{\mu}_i}(z) u_i(z) + \sum_{z \in Z \setminus Z(I)} q_\mu(z) u_i(z). \tag{16}$$

Noticing that the left-hand side of Equation (16) is equal to $\sum_{z \in Z} q_\mu(z) u_i(z)$ and that $y^\sigma_{\mu,\hat{\mu}_i}(z) = p^\sigma_{\mu,\hat{\mu}_i}(z) + q_\mu(z)$, we can rewrite Equation (16) as follows:

$$\sum_{z \in Z(I,a)} q_\mu(z) u_i(z) \geq \sum_{z \in Z(I)} p^\sigma_{\mu,\hat{\mu}_i}(z) u_i(z). \tag{17}$$

A probability distribution $\mu \in \Delta_\Pi$ is said to be an $\epsilon$-EFCE if, for every $i \in \mathcal{P}$ and $(\sigma, \hat{\mu}_i)$-trigger agent for player $i$, with $\sigma = (I, a) \in \Sigma_i$, it holds:

$$\sum_{z \in Z(I,a)} q_\mu(z) u_i(z) \geq \sum_{z \in Z(I)} p^\sigma_{\mu,\hat{\mu}_i}(z) u_i(z) - \epsilon. \tag{18}$$

### A.2  Computation of EFCEs

The problem of computing an optimal EFCE in extensive-form games with more than two players and/or chance moves is known to be NP-hard [32]. However, Huang and von Stengel [19] show that the problem of finding *one* EFCE can be solved in polynomial time via a variation of the *Ellipsoid Against Hope* algorithm [26, 20]. This holds for arbitrary EFGs with multiple players and/or chance moves. Unfortunately, that algorithm is mainly a theoretical tool, and it is known to have limited scalability beyond toy problems. Dudík and Gordon [9] provide an alternative sampling-based algorithm to compute EFCEs. However, their algorithm is *centralized* and based on MCMC sampling which may limit its practical appeal. Our framework is arguably simpler and based on the classical *counterfactual regret minimization* algorithm [34, 12]. Moreover, our framework is fully *decentralized* since each player, at every decision point, plays so as to minimize her internal/external regret.

If we restrict our attention to two-player perfect-recall games without chance moves, than the problem of determining an optimal EFCE can be characterized through a succint linear program with polynomial size in the game description [32]. In this setting, Farina et al. [13] show that the problem of computing an EFCE can be formulated as the solution to a bilinear saddle-point problem, which they solve via a subgradient descent method. Moreover, Farina et al. [14] design a regret minimization algorithm suitable for this specific scenario. In a recent paper, Farina and Sandholm [10] showed that that an optimal EFCE, EFCCE and NFCCE can be computed in polynomial time in the game size in two-player general-sum games that satisfy a condition known as *triangle-freeness*. The triangle-freeness condition holds, for example, when all chance moves are *public*, that is, both players observe all chance moves.

## B  Omitted proofs

### B.1  Proofs for Section 4

The following auxiliary result is exploited in the proof of Theorem 1.

**Lemma 4.** *For every iteration $t = 1, \ldots, T$, player $i \in \mathcal{P}$, plan $\hat{\pi}_i \in \Pi_i$, joint plan $\pi^t = (\pi^t_i, \pi^t_{-i}) \in \Pi$, and infoset $I \in \mathcal{I}_i$, the following holds:*

$$V^t_I(\hat{\pi}_i) - V^t_I(\pi^t) = \sum_{z \in Z(I)} \left( \rho^{(\hat{\pi}_i, \pi^t_{-i})}_{I \to z} - \rho^{\pi^t}_{I \to z} \right) p_c(z) u_i(z).$$

*Proof.* Given an arbitrary infoset $I \in \mathcal{I}_i$, the set of terminal nodes immediately reachable from $I$ through action $a \in A(I)$ is defined as

$$Z^{\mathrm{I}}(I, a) := Z(I, a) \setminus \bigcup_{J \in \mathcal{C}(I,a)} Z(J).$$

By expanding $V_I^t(\hat{\pi}_i)$ according to its definition (Equation (8)) and by substituting the definition of immediate utility vector $u_i^t$ we obtain that

$$
\begin{aligned}
V_I^t(\hat{\pi}_i) &= u_i^t[I, \hat{\pi}_i(I)] + \sum_{J \in \mathcal{C}(I, \hat{\pi}_i(I))} V_J^t(\hat{\pi}_i) \\
&= \sum_{z \in Z^{\mathrm{I}}(I, \hat{\pi}_i(I))} \mathbb{1}[\pi_{-i}^t \in \Pi_{-i}(z)]\, p_c(z) u_i(z) + \sum_{J \in \mathcal{C}(I, \hat{\pi}_i(I))} V_J^t(\hat{\pi}_i) \\
&= \sum_{z \in Z(I)} \rho_{I \to z}^{\hat{\pi}_i} \mathbb{1}[\pi_{-i}^t \in \Pi_{-i}(z)]\, p_c(z) u_i(z),
\end{aligned}
$$

where the last expression is obtained by expanding recursively the terms $V_J^t(\hat{\pi}_i)$. By definition, $\rho_{I \to z}^{(\hat{\pi}_i, \pi_{-i}^t)} = \rho_{I \to z}^{\hat{\pi}_i} \cdot \mathbb{1}[\pi_{-i}^t \in \Pi_{-i}(z)]$. Therefore, we can write $V_I^t(\hat{\pi}_i) = \sum_{z \in Z(I)} \rho_{I \to z}^{(\hat{\pi}_i, \pi_{-i}^t)} p_c(z) u_i(z)$. Analogously, by expanding $V_I^t(\pi_i^t)$, we obtain that $V_I^t(\pi_i^t) = \sum_{z \in Z(I)} \rho_{I \to z}^{\pi^t} p_c(z) u_i(z)$. This concludes the proof. $\square$

**Theorem 1.** *At all times $T$, the empirical frequency of play $\bar{\mu}^T$ (Equation 3) is an $\epsilon$-EFCE, where*

$$\epsilon := \max_{i \in \mathcal{P}} \max_{\sigma \in \Sigma_i} \frac{R_\sigma^T}{T}.$$

*Proof.* Fix any player $i \in \mathcal{P}$ and any sequence $\sigma = (I, a) \in \Sigma_i$ for her. From Lemma 4, the regret $R_\sigma^T$ is

$$
\begin{aligned}
R_\sigma^T &= \max_{\hat{\pi}_i \in \Pi_i(I)} \sum_{t=1}^T \mathbb{1}[\pi_i^t \in \Pi_i(\sigma)] \left( \sum_{z \in Z(I)} \left( \rho_{I \to z}^{(\hat{\pi}_i, \pi_{-i}^t)} - \rho_{I \to z}^{\pi^t} \right) p_c(z) u_i(z) \right) \\
&= \max_{\hat{\pi}_i \in \Pi_i(I)} \sum_{t=1}^T \sum_{\pi \in \Pi} \mathbb{1}[\pi = \pi^t] \left( \mathbb{1}[\pi_i \in \Pi_i(\sigma)] \left( \sum_{z \in Z(I)} \left( \rho_{I \to z}^{(\hat{\pi}_i, \pi_{-i})} - \rho_{I \to z}^{\pi} \right) p_c(z) u_i(z) \right) \right) \\
&= \max_{\hat{\pi}_i \in \Pi_i(I)} \sum_{\pi \in \Pi} \mathbb{1}[\pi_i \in \Pi_i(\sigma)] \left( \left( \sum_{t=1}^T \mathbb{1}[\pi = \pi^t] \right) \left( \sum_{z \in Z(I)} \left( \rho_{I \to z}^{(\hat{\pi}_i, \pi_{-i})} - \rho_{I \to z}^{\pi} \right) p_c(z) u_i(z) \right) \right).
\end{aligned}
$$

By using the definition of empirical frequency of play, we can write $\sum_{t=1}^T \mathbb{1}[\pi = \pi^t] = T \bar{\mu}^T(\pi)$. Hence,

$$
\begin{aligned}
R_\sigma^T &= T \max_{\hat{\pi}_i \in \Pi_i(I)} \sum_{\pi \in \Pi} \mathbb{1}[\pi_i \in \Pi_i(\sigma)] \left( \bar{\mu}^T(\pi) \left( \sum_{z \in Z(I)} \left( \rho_{I \to z}^{(\hat{\pi}_i, \pi_{-i})} - \rho_{I \to z}^{\pi} \right) p_c(z) u_i(z) \right) \right) \\
&= T \max_{\hat{\pi}_i \in \Pi_i(I)} \sum_{\substack{\pi_i \in \Pi_i(\sigma) \\ \pi_{-i} \in \Pi_{-i}}} \bar{\mu}^T(\pi) \left( \sum_{z \in Z(I)} \left( \rho_{I \to z}^{(\hat{\pi}_i, \pi_{-i})} - \rho_{I \to z}^{\pi} \right) p_c(z) u_i(z) \right) \\
&= T \max_{\hat{\pi}_i \in \Pi_i(I)} \sum_{z \in Z(I)} \sum_{\substack{\pi_i \in \Pi_i(\sigma) \\ \pi_{-i} \in \Pi_{-i}}} \bar{\mu}^T(\pi) \left( \rho_{I \to z}^{(\hat{\pi}_i, \pi_{-i})} - \rho_{I \to z}^{\pi} \right) p_c(z) u_i(z).
\end{aligned}
$$

Using the definition of the $\rho_{I \to z}$ symbols, that is,

$$\rho_{I \to z}^{(\hat{\pi}_i, \pi_{-i})} = \rho_{I \to z}^{\hat{\pi}_i} \cdot \mathbb{1}[\pi_{-i} \in \Pi_{-i}(z)], \qquad \rho_{I \to z}^{\pi} = \rho_{I \to z}^{\pi_i} \cdot \mathbb{1}[\pi_{-i} \in \Pi_{-i}(z)],$$

we further obtain

$$R_\sigma^T = T \max_{\hat{\pi}_i \in \Pi_i(I)} \sum_{z \in Z(I)} \sum_{\substack{\pi_i \in \Pi_i(\sigma) \\ \pi_{-i} \in \Pi_{-i}}} \bar{\mu}^T(\pi) \left( \rho_{I \to z}^{\hat{\pi}_i} - \rho_{I \to z}^{\pi_i} \right) \mathbb{1}[\pi_{-i} \in \Pi_{-i}(z)] p_c(z) u_i(z)$$

$$= T \max_{\hat{\pi}_i \in \Pi_i(I)} \sum_{z \in Z(I)} \sum_{\substack{\pi_i \in \Pi_i(\sigma) \\ \pi_{-i} \in \Pi_{-i}(z)}} \bar{\mu}^T(\pi) \left( \rho_{I \to z}^{\hat{\pi}_i} - \rho_{I \to z}^{\pi_i} \right) p_c(z) u_i(z)$$

$$= T \left( \underbrace{\max_{\hat{\pi}_i \in \Pi_i(I)} \sum_{z \in Z(I)} \sum_{\substack{\pi_i \in \Pi_i(\sigma) \\ \pi_{-i} \in \Pi_{-i}(z)}} \bar{\mu}^T(\pi) \rho_{I \to z}^{\hat{\pi}_i} p_c(z) u_i(z)}_{\text{\textcircled{B}}} \right) - T \left( \underbrace{\sum_{z \in Z(I)} \sum_{\substack{\pi_i \in \Pi_i(\sigma) \\ \pi_{-i} \in \Pi_{-i}(z)}} \bar{\mu}^T(\pi) \rho_{I \to z}^{\pi_i} p_c(z) u_i(z)}_{\text{\textcircled{C}}} \right).$$

We now analyze \textcircled{B} and \textcircled{C} separately.

\textcircled{B} By convexity, we have:

$$\text{\textcircled{B}} = \max_{\hat{\mu}_i \in \Delta_{\Pi_i(I)}} \left\{ \sum_{\hat{\pi}_i \in \Pi_i(I)} \hat{\mu}_i(\hat{\pi}_i) \left( \sum_{z \in Z(I)} \sum_{\substack{\pi_i \in \Pi_i(\sigma) \\ \pi_{-i} \in \Pi_{-i}(z)}} \bar{\mu}^T(\pi) \rho_{I \to z}^{\hat{\pi}_i} p_c(z) u_i(z) \right) \right\}$$

$$= \max_{\hat{\mu}_i \in \Delta_{\Pi_i(I)}} \left\{ \sum_{z \in Z(I)} \left( \sum_{\hat{\pi}_i \in \Pi_i(I)} \hat{\mu}_i(\hat{\pi}_i) \rho_{I \to z}^{\hat{\pi}_i} \right) \left( \sum_{\substack{\pi_i \in \Pi_i(\sigma) \\ \pi_{-i} \in \Pi_{-i}(z)}} \bar{\mu}^T(\pi) p_c(z) u_i(z) \right) \right\}.$$

Since $\hat{\pi}_i \in \Pi_i(I)$ and $z \in Z(I)$, $\rho_{I \to z}^{\hat{\pi}_i} = \mathbb{1}[\hat{\pi}_i \in \Pi_i(z)]$. So,

$$\text{\textcircled{B}} = \max_{\hat{\mu}_i \in \Delta_{\Pi_i(I)}} \left\{ \sum_{z \in Z(I)} \left( \sum_{\hat{\pi}_i \in \Pi_i(I)} \hat{\mu}_i(\hat{\pi}_i) \mathbb{1}[\hat{\pi}_i \in \Pi_i(z)] \right) \left( \sum_{\substack{\pi_i \in \Pi_i(\sigma) \\ \pi_{-i} \in \Pi_{-i}(z)}} \bar{\mu}^T(\pi) p_c(z) u_i(z) \right) \right\}$$

$$= \max_{\hat{\mu}_i \in \Delta_{\Pi_i(I)}} \left\{ \sum_{z \in Z(I)} \left( \sum_{\hat{\pi}_i \in \Pi_i(z)} \hat{\mu}_i(\hat{\pi}_i) \right) \left( \sum_{\substack{\pi_i \in \Pi_i(\sigma) \\ \pi_{-i} \in \Pi_{-i}(z)}} \bar{\mu}^T(\pi) \right) p_c(z) u_i(z) \right\}$$

$$= \max_{\hat{\mu}_i \in \Delta_{\Pi_i(I)}} \left\{ \sum_{z \in Z(I)} p_{\bar{\mu}^T, \hat{\mu}_i}^{\sigma}(z) \, u_i(z) \right\}. \tag{19}$$

\textcircled{C} Since $\pi_i \in \Pi_i(\sigma) \subseteq \Pi_i(I)$ and $z \in Z(I)$, $\rho_{I \to z}^{\pi_i} = \mathbb{1}[z \in Z(\sigma)] \cdot \mathbb{1}[\pi_i \in \Pi_i(z)]$. Therefore,

$$\text{\textcircled{C}} = \sum_{z \in Z(I)} \sum_{\substack{\pi_i \in \Pi_i(\sigma) \\ \pi_{-i} \in \Pi_{-i}(z)}} \bar{\mu}^T(\pi) \, \mathbb{1}[z \in Z(\sigma)] \mathbb{1}[\pi_i \in \Pi_i(z)] \, p_c(z) u_i(z)$$

$$= \sum_{z \in Z(I)} \left( \mathbb{1}[z \in Z(\sigma)] \sum_{\substack{\pi_i \in \Pi_i(\sigma) \\ \pi_{-i} \in \Pi_{-i}(z)}} \bar{\mu}^T(\pi) \, \mathbb{1}[\pi_i \in \Pi_i(z)] \, p_c(z) u_i(z) \right)$$

$$= \sum_{z \in Z(\sigma)} \left( \sum_{\substack{\pi_i \in \Pi_i(z) \\ \pi_{-i} \in \Pi_{-i}(z)}} \bar{\mu}^T(\pi) \right) p_c(z) u_i(z)$$

$$= \sum_{z \in Z(\sigma)} q_{\bar{\mu}^T}(z) \, u_i(z). \tag{20}$$

Substituting the expressions in (19) and (20) into the expression for $R_\sigma^T$, we obtain

$$\frac{R_\sigma^T}{T} = \max_{\hat{\mu}_i \in \Delta_{\Pi_i(I)}} \left\{ \sum_{z \in Z(I)} p_{\bar{\mu}^T, \hat{\mu}_i}^{\sigma}(z) \, u_i(z) \right\} - \sum_{z \in Z(\sigma)} q_{\bar{\mu}^T}(z) \, u_i(z). \tag{21}$$

Finally, using the hypothesis, we can write

$$\epsilon = \max_{i\in\mathcal{P}} \max_{\sigma\in\Sigma_i} \frac{R_\sigma^T}{T}$$

$$= \max_{i\in\mathcal{P}} \max_{\sigma\in\Sigma_i} \left\{ \max_{\hat{\mu}_i\in\Delta_{\Pi_i(I)}} \left\{ \sum_{z\in Z(I)} p_{\bar{\mu}^T,\hat{\mu}_i}^\sigma(z)\, u_i(z) \right\} - \sum_{z\in Z(\sigma)} q_{\bar{\mu}^T}(z)\, u_i(z) \right\}$$

$$= \delta(\bar{\mu}^T).$$

This concludes the proof. $\qquad\square$

**Corollary 1.** *If* $\limsup_{T\to\infty} \max_{i\in\mathcal{P}} \max_{\sigma\in\Sigma_i} \dfrac{R_\sigma^T}{T} \leq 0$, *then* $\limsup_{T\to\infty} \delta(\bar{\mu}^T) \leq 0$, *that is, for any* $\epsilon > 0$, *eventually the empirical frequency of play* $\bar{\mu}^T$ *becomes an* $\epsilon$-*EFCE.*

*Proof.* By Equation (21) we obtain:

$$0 \geq \limsup_{T\to\infty} \frac{R_\sigma^T}{T}$$

$$= \limsup_{T\to\infty} \left( \max_{\hat{\mu}_i\in\Delta_{\Pi_i(I)}} \left\{ \sum_{z\in Z(I)} p_{\bar{\mu}^T,\hat{\mu}_i}^\sigma(z)\, u_i(z) \right\} - \sum_{z\in Z(\sigma)} q_{\bar{\mu}^T}(z)\, u_i(z) \right)$$

$$\geq \max_{\hat{\mu}_i\in\Delta_{\Pi_i(I)}} \left\{ \limsup_{T\to\infty} \sum_{z\in Z(I)} p_{\bar{\mu}^T,\hat{\mu}_i}^\sigma(z)\, u_i(z) \right\} - \limsup_{T\to\infty} \sum_{z\in Z(\sigma)} q_{\bar{\mu}^T}(z)\, u_i(z),$$

where the last inequality follows from swapping the order of $\limsup$ and $\max$. By definition of $\limsup$, for any $\epsilon > 0$, eventually $\delta(\bar{\mu}^T) < 0$ (more precisely: for any $\epsilon > 0$, there must be a $\tau = \tau(\epsilon)$ such that $\delta(\bar{\mu}^T) < \epsilon$ for all $T \geq \tau$), which means that eventually the empirical frequency of play $\bar{\mu}^T$ becomes an $\epsilon$-EFCE. $\qquad\square$

## B.2 Proof for Section 5

**Lemma 1.** *The subtree regret for each player* $i \in \mathcal{P}$, *sequence* $\sigma = (J, a') \in \Sigma_i$, *and infoset* $I \in \mathcal{C}^\star(J)$ *can be decomposed as:*

$$R_{\sigma,I}^T = \max_{a\in A(I)} \left\{ \hat{R}_{\sigma,I,a}^T + \sum_{I'\in\mathcal{C}(I,a)} R_{\sigma,I'}^T \right\}.$$

*Proof.* By using the recursive definitions of $R_{\sigma,I}^T$ and $V_I^t(\hat{\pi}_i)$, we get:

$$R_{\sigma,I}^T = \max_{\hat{\pi}_i\in\Pi_i(I)} \left\{ \sum_{t=1}^T \mathbb{1}[\pi_i^t \in \Pi_i(\sigma)]\big(V_I^t(\hat{\pi}_i) - V_I^t(\pi_i^t)\big) \right\}$$

$$= \max_{\hat{\pi}_i\in\Pi_i(I)} \left\{ \sum_{t=1}^T \mathbb{1}[\pi_i^t \in \Pi_i(\sigma)]V_I^t(\hat{\pi}_i) \right\} - \sum_{t=1}^T \mathbb{1}[\pi_i^t \in \Pi_i(\sigma)]V_I^t(\pi_i^t)$$

$$= \max_{\hat{\pi}_i\in\Pi_i(I)} \left\{ \sum_{t=1}^T \mathbb{1}[\pi_i^t \in \Pi_i(\sigma)]\left( u_i^t[I,\hat{\pi}_i(I)] + \sum_{I'\in\mathcal{C}(I,\hat{\pi}_i(I))} V_{I'}^t(\hat{\pi}_i) \right) \right\}$$

$$\qquad\qquad\qquad\qquad - \sum_{t=1}^T \mathbb{1}[\pi_i^t \in \Pi_i(\sigma)]V_I^t(\pi_i^t)$$

$$= \max_{a\in A(I)} \left\{ \sum_{t=1}^T \mathbb{1}[\pi_i^t \in \Pi_i(\sigma)]u_i^t[I,a] + \sum_{I'\in\mathcal{C}(I,a)} \max_{\hat{\pi}_i\in\Pi_i(I')} \left\{ \sum_{t=1}^T \mathbb{1}[\pi_i^t \in \Pi_i(\sigma)]V_{I'}^t(\hat{\pi}_i) \right\} \right\}$$

$$-\sum_{t=1}^{T}\mathbb{1}[\pi_i^t \in \Pi_i(\sigma)]V_I^t(\pi_i^t)$$

$$= \max_{a\in A(I)}\left\{\sum_{t=1}^{T}\mathbb{1}[\pi_i^t \in \Pi_i(\sigma)]u_i^t[I,a] + \sum_{I'\in\mathcal{C}(I,a)}\left(R_{\sigma,I'}^T + \sum_{t=1}^{T}\mathbb{1}[\pi_i^t \in \Pi_i(\sigma)]V_{I'}^t(\pi_i^t)\right)\right\}$$

$$-\sum_{t=1}^{T}\mathbb{1}[\pi_i^t \in \Pi_i(\sigma)]V_I^t(\pi_i^t),$$

where the last step is by definition of subtree regret. By rewriting the above expression according to Equation (9) we get the result. $\square$

**Lemma 2.** *For every player $i \in \mathcal{P}$, sequence $\sigma = (J, a') \in \Sigma_i$, and infoset $I \in \mathcal{C}^\star(J)$, it holds:*

$$R_{\sigma,I}^T \le \max_{\hat{\pi}_i\in\Pi_i(I)}\sum_{I'\in\mathcal{C}^\star(I)}\mathbb{1}[\hat{\pi}_i \in \Pi_i(I')]\,\hat{R}_{\sigma,I'}^T. \tag{12}$$

*Proof.* Consider an arbitrary sequence $\sigma = (J, a') \in \Sigma_i$ and infoset $I \in \mathcal{C}^\star(J)$. By Lemma 1 we have:

$$R_{\sigma,I}^T = \max_{a\in A(I)}\left\{\sum_{t=1}^{T}\mathbb{1}[\pi_i^t \in \Pi_i(\sigma)]\big(\hat{u}_I^t(a) - \hat{u}_I^t(\pi_i^t(I))\big) + \sum_{I'\in\mathcal{C}(I,a)}R_{\sigma,I'}^T\right\}$$

$$= \max_{a\in A(I)}\left\{\sum_{t=1}^{T}\mathbb{1}[\pi_i^t \in \Pi_i(\sigma)]\hat{u}_I^t(a) + \sum_{I'\in\mathcal{C}(I,a)}R_{\sigma,I'}^T\right\} - \sum_{t=1}^{T}\mathbb{1}[\pi_i^t \in \Pi_i(\sigma)]\hat{u}_I^t(\pi_i^t(I))$$

$$\le \max_{a\in A(I)}\left\{\sum_{t=1}^{T}\mathbb{1}[\pi_i^t \in \Pi_i(\sigma)]\hat{u}_I^t(a)\right\} + \max_{a\in A(I)}\left\{\sum_{I'\in\mathcal{C}(I,a)}R_{\sigma,I'}^T\right\} - \sum_{t=1}^{T}\mathbb{1}[\pi_i^t \in \Pi_i(\sigma)]\hat{u}_I^t(\pi_i^t(I))$$

$$= \hat{R}_{\sigma,I}^T + \max_{a\in A(I)}\left\{\sum_{I'\in\mathcal{C}(I,a)}R_{\sigma,I'}^T\right\}.$$

By starting from $I$ and applying the above equation inductively, we obtain the result. $\square$

### B.3 Proofs for Section 6

**Lemma 5.** *For any $I \in \mathcal{I}_i$ and $t = 1, \ldots, T$, if it is the case that $\pi_i^t \notin \Pi_i(I)$, then the sequence $\sigma_I^t$ defined by* SampleInternal *exists and is unique.*

*Proof.* It is enough to proceed from infoset $I$ towards the root of the tree. Eventually, the procedure reaches an infoset $I' \in \mathcal{I}_i$ such that $\pi_i^t \in \Pi_i(I')$. Then, $\sigma_I^t$ is identified by the pair $(I', \pi_i^t(I'))$. $\square$

**Lemma 3.** *For any $I, J \in \mathcal{I}_i : I \preceq J$, if $\hat{R}_{\sigma,J}^T = o(T)$ for all $\sigma = (I, a) \in \Sigma_i$ then $\hat{R}_{\sigma(I),J}^T = o(T)$.*

*Proof.* By hypothesis and since the action space $A(I)$ is finite we have that

$$\sum_{a\in A(I)}\hat{R}_{(I,a),J}^T = o(T).$$

Moreover,

$$\sum_{a\in A(I)}\hat{R}_{(I,a),J}^T = \sum_{a\in A(I)}\max_{\hat{a}\in A(J)}\left\{\sum_{t=1}^{T}\mathbb{1}[\pi_i^t \in \Pi_i(I,a)]\big(\hat{u}_J^t(\hat{a}) - \hat{u}_J^t(\pi_i^t(J))\big)\right\}$$

$$\ge \max_{\hat{a}\in A(J)}\left\{\sum_{t=1}^{T}\sum_{a\in A(I)}\mathbb{1}[\pi_i^t \in \Pi_i(I,a)]\big(\hat{u}_J^t(\hat{a}) - \hat{u}_J^t(\pi_i^t(J))\big)\right\}$$

$$= \max_{\hat{a} \in A(J)} \left\{ \sum_{t=1}^{T} \mathbb{1}[\pi_i^t \in \Pi_i(I)] \left( \hat{u}_J^t(\hat{a}) - \hat{u}_J^t(\pi_i^t(J)) \right) \right\}$$

$$= \max_{\hat{a} \in A(J)} \left\{ \sum_{t=1}^{T} \mathbb{1}[\pi_i^t \in \Pi_i(\sigma(I))] \left( \hat{u}_J^t(\hat{a}) - \hat{u}_J^t(\pi_i^t(J)) \right) \right\}$$

$$= \hat{R}_{\sigma(I),J}^T.$$

This concludes the proof. □

**Theorem 2.** *When all the players play according to* ICFR, *$\bar{\mu}^T$ converges almost surely to an EFCE.*

*Proof.* By Theorem 1, in order to converge to an EFCE, it is enough to minimize the trigger regrets $R_{\sigma}^T$ for each player $i \in \mathcal{P}$ and sequence $\sigma = (I,a) \in \Sigma_i$. This can be done by minimizing the subtree regrets $R_{\sigma,I}^T$ via the minimization of laminar subtree regrets $\hat{R}_{\sigma,I}^T$ for each sequence $\sigma = (J,a) \in \Sigma_i$ and infoset $I \in \mathcal{C}^\star(J)$ (Lemma 2).

For any infoset $I \in \mathcal{I}_i$, the laminar subtree regrets $\hat{R}_{\sigma,I}^T$ are partitioned in three groups on the basis of the trigger sequence $\sigma$:

- *Group 1*: $\sigma = (I,a) \in \Sigma_i$. Laminar subtree regrets belonging to this group are updated at rounds $t$ such that $\pi_i^t \in \Pi_i(I)$, otherwise they remain unchanged. Therefore, they are only updated when the strategy at $I$ is recommended by the internal-regret minimizer $\mathcal{R}_I^{\text{INT}}$, which guarantees $\hat{R}_{\sigma,I}^T = o(T)$ [6].

- *Group 2*: $\sigma = (J,a) \in \Sigma_i$ is such that $J \preceq I$, $J \neq I$, and $a$ is *not* on the path from $J$ to $I$ (*i.e.*, for any $\pi_i \in \Pi_i$, $\pi_i(J) = a$ implies $\pi_i \notin \Pi_i(I)$). The sequence $\sigma_I^t$ is defined as a sequence compatible with $\pi_i^t$ and belonging to $\Sigma_i^c(I)$. By Lemma 5, for each $I \in \mathcal{I}_i$ and $t = 1, \dots, T$, $\sigma_I^t$ exists and is unique. Then, at most one laminar subtree regret term of Group 2 is updated at each round $t$, otherwise they are left unchanged. Whenever one of these regrets is affected by the choice at $t$, the action at $I$ is selected according to the external-regret minimizer $\mathcal{R}_{\sigma_I^t,I}^{\text{EXT}}$. This ensures that each laminar subtree regret belonging to this group is $o(T)$ by the known properties of no-external-regret algorithms [6].

- *Group 3*: $\sigma = (J,a) \in \Sigma_i$ is such that $J \preceq I$, $J \neq I$, and $a$ is on the path from $J$ to $I$ (notice that for each $J \preceq I$, $J \neq I$ one such $a$ is unique because player $i$ has perfect recall). Let $I' \in \mathcal{I}_i$ be such that $J \preceq I' \preceq I$ and $I' \in \mathcal{C}(J,a)$. Notice that, given $I$, $J$, and $\sigma$, one such $I'$ is unique because of the perfect recall assumption. By Lemma 3, we know that if $\hat{R}_{\sigma',I}^T = o(T)$ for all $\sigma' = (I',a') \in \Sigma_i$, then it must be the case that $\hat{R}_{\sigma,I}^T = o(T)$ (notice that $\sigma = (J,a)$ is the same as $\sigma(I')$). By applying the lemma recursively, until all $\sigma'$ belong to either Group 1 or 2, we can guarantee that $\hat{R}_{\sigma,I}^T = o(T)$.

This concludes the proof. □

## C  Experimental Evaluation

Appendix C.1 provides a detailed description of the benchmark games used in our experiments. Finally, Appendix C.2 shows additional experimental results for ICFR.

### C.1  Benchmark games

The size (in terms on number of infosets and sequences) of the parametric instances we use as benchmark is described in Figure 4. In the following, we provide a detailed explanation of the rules of the games.

| | $|\mathcal{P}|$ | Ranks | Player | Infosets | Sequences |
|---|---|---|---|---|---|
| Kuhn | 3 | 3 | Player 1 | 12 | 25 |
| | | | Player 2 | 12 | 25 |
| | | | Player 3 | 12 | 25 |
| | 3 | 4 | Player 1 | 16 | 33 |
| | | | Player 2 | 16 | 33 |
| | | | Player 3 | 16 | 33 |
| Goofspiel | 2 | 3 | Player 1 | 213 | 262 |
| | | | Player 2 | 213 | 262 |
| | 2 | 4 | Player 1 | 8716 | 10649 |
| | | | Player 2 | 8716 | 10649 |
| | 3 | 3 | Player 1 | 837 | 934 |
| | | | Player 2 | 837 | 934 |
| | | | Player 3 | 837 | 934 |
| Leduc | 3 | 3 | Player 1 | 3294 | 7687 |
| | | | Player 2 | 3294 | 7687 |
| | | | Player 2 | 3294 | 7687 |

| | Grid | Rounds | Player | Infosets | Sequences |
|---|---|---|---|---|---|
| Battleship | $(2, 2)$ | 3 | Player 1 | 1413 | 2965 |
| | | | Player 2 | 1873 | 4101 |

Figure 4: The size of our parametric game instances in terms of number of sequences and infosets for each player of the game.

**Kuhn poker**　　The two-player version of the game was originally proposed by [22], while the three-player variation is due to [11]. In a three-player Kuhn poker game with rank $r$, there are $r$ possible cards. Each player initially pays one chip to the pot, and she/he is dealt a single private card. The first player may *check* or *bet* (*i.e.,* put an additional chip in the pot). Then, the second player can check or bet after a first player's check, or *fold/call* the first player's bet. If no bet was previously made, the third player can either check or bet. Otherwise, she/he has to fold or call. After a bet of the second player (resp., third player), the first player (resp., the first and the second players) still has to decide whether to fold or to call the bet. At the showdown, the player with the highest card who has not folded wins all the chips in the pot.

**Goofspiel**　　This game was originally introduced by [27]. Goofspiel is essentially a bidding game where each player has a hand of cards numbered from 1 to $r$ (*i.e.*, the rank of the game). A third stack of $r$ cards is shuffled and singled out as prizes. Each turn, a prize card is revealed, and each player privately chooses one of her/his cards to bid, with the highest card winning the current prize. In case of a tie, the prize card is discarded. After $r$ turns, all the prizes have been dealt out and the payoff of each player is computed as follows: each prize card's value is equal to its face value and the players' scores are computed as the sum of the values of the prize cards they have won. We remark that due to the tie-breaking rule that we employ, even two-player instances of the game are general-sum. All the Goofspiel instances have *limited information*, *i.e.*, actions of the other players are observed only at the end of the game. This makes the game strategically more challenging, as players have less information regarding previous opponents' actions.

**Leduc**　　We use a three-player version of the classical Leduc hold'em poker introduced by Southey et al. [30]. In a Leduc game instance with $r$ ranks the deck consists of three suits with $r$ cards each. As the game starts players pay one chip to the pot. There are two betting rounds. In the first one a single private card is dealt to each player while in the second round a single board card is revealed. The maximum number of raise per round is set to two, with raise amounts of 2 and 4 in the first and second round, respectively.

Battleship is a parametric version of the classic board game, where two competing fleets take turns at shooting at each other. For a detailed explanation of the Battleship game see the work by [13] that introduced it. Our instance has loss multiplier equal to 2, and one ship of length 2 and value 1 for each player

## C.2　Additional results

We provide detailed results on the convergence of ICFR in terms of players' incentives to deviate from the obtained empirical frequency of play $\bar{\mu}^T$. For EFCEs, these incentives correspond to the maximum deviation of each player,

as defined in the outer maximization in Equation (7). Intuitively, for each player, this represents the maximum utility any trigger agent for that player could gain by deviating from the point in which it gets triggered onwards.

We do not only consider deviations as prescribed by EFCE, but we also show results for other solutions concepts involving correlation in EFGs, namely EFCCEs and NFCCEs (see Appendix A for their informal description, while for their formal definitions the reader can refer to Farina et al. [15]). As for EFCCEs, the players' incentives to deviate are defined in a way similar to EFCE, using the definition of trigger agent suitable for EFCCEs (see [15]). Instead, for NFCCEs, each player's incentive to deviate corresponds to the utility she/he could gain by playing the best normal-form plan given $\bar{\mu}^T$. We recall that the following relation holds: EFCE $\subseteq$ EFCCE $\subseteq$ NFCCE.

In Figures 5 - 11, we report players' incentives to deviate obtained with ICFR for EFCE (Left), EFCCE (Center), and NFCCE (Right). As the plots show, the convergence rate is similar for the three cases, electing ICFR as an appealing algorithm also for EFCCEs and NFCCEs. This is the first example of algorithm computing $\epsilon$-EFCCEs efficiently in general-sum EFGs with more than two players and chance. Celli et al. [5] propose some algorithms to compute $\epsilon$-NFCCEs in general-sum EFGs with an arbitrary number of players (including chance). Our algorithm outperforms those of Celli et al. [5], since the latter cannot reach a $0.1$-NFCCE in less than 24h on a Leduc instance with 1200 total infosets and a one-bet maximum per bidding round. Instead, ICFR reaches $\epsilon = 0.1$ in around 9h on an arguably more complex Leduc instance (*i.e.*, more than 9k total infosets and a two-bet maximum per round).

Figure 5: Players' incentives to deviate with ICFR in the Battleship game.

Figure 6: Players' incentives to deviate with ICFR in two-player Goofspiel with 3 ranks.

Figure 7: Players' incentives to deviate with ICFR in two-player Goofspiel with 4 ranks.

Figure 8: Players' incentives to deviate with ICFR in three-player Goofspiel with 3 ranks.

Figure 9: Players' incentives to deviate with ICFR in three-player Kuhn Poker with 3 ranks.

Figure 10: Players' incentives to deviate with ICFR in three-player Kuhn Poker with 4 ranks.

Figure 11: Players' incentives to deviate with ICFR in three-player Leduc Poker with 3 ranks.