[Reviews · NeurIPS 2020]

Review 1

Summary and Contributions: The authors provide a regret-minimisation approach to computing an analogue to correlated equilibria in extensive form games called extensive-form correlated equilibria (EFCE). It was previously unknown whether EFCE can be achieved via uncoupled no-regret dynamics as in typical correlated equilibria in simultaneous games, and the authors provide a way of doing so by introducing an appropriate notion of regret in the extensive form setting (that lines up with the notion of approximation in approximate EFCE), and demonstrating how achieving low regret in this setting suffices to have an approximate EFCE for joint strategy profiles that arise from empirical frequencies of play. As mentioned before, the relevant notion of equilibrium in this setting are extensive-form correlated equilibria (EFCE). Such an equilibrium is a joint distribution over the space of all possible plans of all agents. As is typical in an extensive-form setting, a plan is simply a mapping from information sets to their relevant action profiles that dictates what an agent does at any given situation of play. In the simultaneous setting, a mixed strategy profile is a correlated equilibrium when no agent wishes to deviate from the joint strategy profile, conditional on their realised strategy profile and prior knowledge of the joint distribution of play. In the extensive-form case, deviations from the correlated signal to all agents (which defines a plan each agent takes), correspond to "trigger agents", which correspond to following the profile sampled from a joint distribution "up to" an arbitrary information set, and henceforth deviating via any given plan at future information sets. A joint plan is thus an EFCE if no agent has a beneficial trigger at any information set. A joint strategy profile is an \epsilon EFCE if the benefit from such triggering is at most epsilon for any given player at any given information set. The authors proceed to define a natural version of regret in this setting, called "trigger regret", which measures how much an agent could have benefited from triggering a deviation at a prescribed information set over an observed sequence of plans (i.e. playing the overall sequential game multiple times). Their first main result is showing that if a joint plan is created from the empirical frequency of play over a given time horizon, the approximation factor of an \epsilon EFCE can be bounded by the largest trigger regret of an agent, hence no-regret algorithms in this setting converge to EFCE. Finally, the authors give a no-regret algorithm for repeated play, which rests upon bounding trigger regret with an appropriate local regret term (subtree regret), which is akin to regret conditional upon reaching a given information set in play. The authors' main algorithm bounds this latter quantity via local regret minimisation algorithms at each information set. Each turn, an agents plan consists of giving an action per information set. To construct this, the algorithm sees whether the information set could be reached with the previous turns plan, in which case it performs an internal regret minimisation step, otherwise it performs an external regret minimisation step (minimising swap regret). This algorithm converges almost surely to EFCE. Finally, the authors provide experimental validation of their results, computing epsilon-EFCE for various game instances, demonstrating an empirical convergence rate of O(1/T).

Strengths: I think this was a strong paper, which did a very good job of presenting the relevant results. As they mention in their introduction, the fact that correlated equilibria can be computed efficiently in a completely decentralised fashion via no-regret algorithms is a fundamental result in algorithm equilibrium computation, and generalising this to extensive-form games is an important result. Furthermore, working with extensive-form games is always notationally dense, and the authors have done a great job of explaining things with proper intuition throughout the paper. The authors have also addressed reviewer points in the rebuttal phase well. My score remains the same.

Weaknesses: In spite of the fact that notation is handled well, I wonder if there is any scope to include some proofs of theoretical results within the paper, or perhaps give intuition to the proofs. I would have liked to see some comments on the technical approach to showing that minimising average trigger regret results in EFCE in the limit. Though the definition is very natural, and the result is not surprising, it would be interesting to know if there are hurdles different from the simultaneous action case. I also wonder whether the authors have explored the convergence rate of the algorithm from a theoretical angle.

Correctness: To my understanding, the claims and empirical methodology seem correct.

Clarity: I believe the paper is very well written.

Relation to Prior Work: The authors have given a clear connection to how regret minimisation ties into computing correlated equilibria in simultaneous games. P

Reproducibility: Yes

Additional Feedback: Throughout most of the paper the authors give good intuition to definitions, lemmas, etc. The one place I thought this was lacking was in Lemma 1 and 2, where subtree regret is defined and it is shown that this is a bound to trigger regret. A simple sentence after the fact would be great to ease the readability.


Review 2

Summary and Contributions: This paper attempts to present an online learning algorithm for extensive-form games where its empirical play in self-play converges to an extensive-form correlated equilibrium. The algorithm, internal counterfactual regret minimization (ICFR), does not have the social welfare guarantees of previous algorithms and does not optimize this criterion in practice either, but ICFR is also substantially faster, more scalable, and more general. To derive this algorithm, the authors introduce the idea of trigger regret to connect EFCE deviation incentives to online learning, show that the previously developed extensive-form regret decompositions apply to trigger regret, and suggest an algorithm built on the counterfactual regret minimization (CFR) framework.

Strengths: The paper sets up the problem and its analytical frameworks well. It also presents a clever way to combine regret minimizers with a sampling scheme. The theoretical results are straightforward but worthwhile. The experimental results show the new algorithm performs well.

Weaknesses: *Update after rebuttal* The rebuttal correctly pointed out that I was mistaken, Sigma_i^c and line 18 of the ICFR algorithm are well defined. I got confused by the multiple uses of sigma and missed that overloads were used in those places, but I think that was my fault, it's clear to me now. The fact that "all but one" should read "only one" in one of the sentences describing the update procedure also makes the algorithm make much more sense to me now. I still think the algorithm could be described more clearly, particuarly the update procedure, but I am satisfied that the rebuttal confirmed that this would be worked on for the final draft. *Before rebuttal* This paper's greatest weakness is its lack of clarify in describing the ICFR algorithm, which is described in the "Clarity" section of this review. Beyond clarity, there are a few minor weaknesses in the results: 1. The algorithm and analysis is based on regret minimization, which typically facilitates a finite time performance bound, but all of the theoretical results are stated and proven in terms of asymptotic convergence in the limit. So the tools being used would suggest stronger results than those provided. 2. It is stated that ICFR is much more scalable than alternative algorithms, but there is no clear summary and accounting of the computational requirements for ICFR compared to its peers. 3. Roughly five of this paper's eight pages are used as setup for the paper's contributions. This leaves little space to discuss the main algorithm, experiments, and does not leave any room for a concluding section. While much of this setup is instructive, it might improve the paper to tradeoff some of the setup in return for a deeper discussion of the main contributions and their implications.

Correctness: *Update after rebuttal* I am more confident now that all claims are correct. *Before rebuttal* The results leading up to Section 6 appear correct. However, the lack of clarity in the description of the ICFR algorithm makes assessing the correctness of this central contribution difficult. The algorithm I can infer does appear to be correct though and the experimental results reflect this. The empirical methodology appears to be sound and appropriate.

Clarity: Everything except most of Section 6 is presented clearly, where Section 6 presents the ICFR algorithm. However, because this section describes the main contribution of the paper, the lack of clarity in this particular section is a major concern. It can maybe be addressed for the final version of the paper, but as is, the ICFR algorithm is not fully specified, which is why I have recommended that this paper is marginally below the acceptance threshold and why it might be unduly difficult to reproduce. The idea of sampling different portions of the plans for each round according to different regret minimizers according to whether the information set is on or off the line of play is clear. And it is clear that down the line of play, internal regret minimizers associated with each trigger sequence is queried. But the instantiation of external regret minimizers is confusing. The wording of the English description (the sentence spanning lines 237-242) is difficult to parse. In the following sentence, the set Sigma^c_i(I) does not appear to be well defined as the symbol sigma is undefined. This is less of a problem for the sampling procedure, where the SampleInternal function of Algorithm 1 and the rest of the paragraph provides enough context to potentially infer the sampling algorithm. But the UpdateInternal function is defined in terms of Sigma^c_i(I) in addition to Pi(sigma(I)) where sigma is again undefined. I am unable to understand the update procedure from the English paragraph from lines 252-256 either. The best guess I can make about the update procedure is that along the line of play, the internal regret minimizers are updated using the obvious instantaneous trigger regrets and off the line of play, all external regret minimizers corresponding to each possible trigger prefix along the line of play. The correctness of Theorem 2 hinges on the the correctness of the update procedure. I think that the procedure I have inferred makes this theorem true, but it contradicts the statement that ". . . all but one of the regret minimizers will receive a non-zero utility" because this is only true for the information sets off the line of play, otherwise only the internal regret minimizer will be updated. The example also does not describe the regret update procedure. A final minor point, but naming the ICFR subroutines with the "Internal" postfix is somewhat confusing because external regret minimizers are also sampled or updated therein. The same criticism could be levied at the naming "Internal CFR" as well. Since Dudík and Gordon (2009) describes the deviations involved in EFCEs "causal", perhaps this terminology would be better, and the algorithm could be something like "causal pure CFR". References ========== Dudík, M. and G. J. Gordon (2009). "A Sampling-Based Approach to Computing Equilibria in Succinct Extensive-Form Games". In: Proceedings of the 25th Conference on Uncertainty in Artificial Intelligence (UAI-2009).

Relation to Prior Work: Discussions of prior work is appropriate and largely complete. The discussion of background concepts and work is helpful and clear. However, there are two items where the discussion of prior work could be improved: Section 5 describes how the laminar regret decomposition is repurposed for trigger regret. However, the laminar regret decomposition is a generalization of the counterfactual regret decomposition introduced by Zinkevich (2008) to convex losses and compact convex decision sets, neither of which are required by any of the results in Section 5 since the decision sets are discrete. So while it is not incorrect to use this terminology, it is less precise and create some confusion. Regardless of whether the terminology is changed, it would be worthwhile to include a citation to Zinkevich (2008) to reference this connection. Similarly, it would be worth pointing out that hat(u)^t_I (9) is the counterfactual value function of I wrt pi^t. Because each of the strategies employed by ICFR on each iteration is a pure normal-form plan, the ICFR algorithm is actually an instance of pure CFR (Gibson, 2014). Connecting to this algorithm could allow the derivation of finite time regret and equilibrium approximation bounds, as well as help readers to understand the algorithm better. References ========== Gibson, Richard G. "Regret minimization in games and the development of champion multiplayer computer poker-playing agents." (2014).

Reproducibility: Yes

Additional Feedback: 1. Should (7) be <= epsilon? Conventionally an epsilon'-equilibrium, epsilon' < epsilon, is also an epsilon-equilibrium. 2. It looks like the first summation in Lemma 1 could be rewritten as hat{R}^T_{sigma, I, a}, and this would make it easier to read.


Review 3

Summary and Contributions: Post-response: My opinion remain unchanged after the author response and reviewer discussions. This paper introduces the internal counterfactual regret minimization algorithm (ICFR). It shows that when all players of an perfect recall extensive-form game play according to ICFR that their empirical joint-distribution of play converges to an extensive-form correlated equilibrium (EFCE). This is a significant results for a number of reasons. It is simpler and more scalable than prior approaches for general extensive-form games. e.g., the Ellipsoid against Hope algorithm and its variants, and the method of Dudik and Gordon, are complicated and computationally expensive. Both methods are also "central" in that a single computation computes the EFCE for all the agents to follow. More scalable methods, like the one presented in Farina 2019, are not applicable in games with more than two players or with chance moves. ICFR is scalable, relatively simple, decentralized, and applicable in all games. Achieving the latter two properties alone is a long-standing open problem that spans computer science, economics and game theory. This year's SIGecon test of time award winners introduced decentralized learning algorithms that converge to correlated equilibria in normal-form games.

Strengths: As stated above, ICFR is a significant advancement over prior work that is highly relevant to the NeurIPS community. It is novel, intuitive and fairly understandable.

Weaknesses: There is one potential drawback that I see to ICFR over prior work that is not discussed in the paper. An arbitrary EFCE cannot be stored in general since there are an exponential number of normal-form plans. Therefore, one essentially must record the sampled normal-form plans over the course of the algorithm to "output" the EFCE. This drawback is shared by the Ellipsoid against Hope and Dudik and Gordon's approach, but both of those methods have convergence guarantees, which bounds the number of normal-form plans necessary. The experimental results provide some reassurance that this not a huge problem, but some theoretical justification or a procedure to avoid recording all iterates would be very useful. Said another way, it has not been demonstrated that ICFR is a strict win over prior approaches. It is worth noting this drawback in the text. Though the scalability is a lot better than prior algorithms, to be fair it is still an issue that makes ICFR impractical to use beyond small games. e.g., the paper states that ICFR takes 9 hours to reach a gap of 0.1 in a game with 9k information sets. To nitpick, the experiments will not be easy to reproduce for most readers. ICFR itself looks fairly easy to implement, but, e.g., computing the EFCE gap itself is non-trivial.

Correctness: The algorithm appears to be correct and I have checked the proofs fairly closely.

Clarity: Yes.

Relation to Prior Work: Yes.

Reproducibility: Yes

Additional Feedback: Great work! Do you have any idea how correlated the play that ICFR converges to in the experiments? and how much "better" the play is compared to using simple external regret minimizers?

[Author Response · NeurIPS 2020]

**Reviewer 1:** Thanks! We will include some intuition for Lemmas 1 and 2. If the paper is accepted, we will include additional details on the proofs with the ninth content page for the camera-ready version. An accurate theoretical analysis of the convergence rate is left as a future development of this work. The present paper presents a framework that can work with arbitrary external and internal regret minimizers. The convergence rate will definitely be impacted by the specific choice of such regret minimizers.

**Reviewer 2:** Thanks for your feedback!
— Re "*theoretical results are straightforward but worthwhile*". We strongly disagree that the results are straightforward. Deriving no-regret dynamics for correlated equilibria in extensive-form games has been a challenging open problem for years. Related work trying to solve the same problem date back to the early '00s, which may be an indicator that the solution to the problem was not obvious.
— Re "*finite-time analysis*". We leave giving sharp finite-time bounds as an open future direction.
— Re clarity. Thanks for your feedback! We will expand the description of the algorithm and the example in the final version, using the ninth page of content.
— Re "*set $\Sigma_i^c(I)$...is undefined*". We are not sure which symbol $\Sigma$ you were referring to. The symbol $\Sigma_i^c(I)$ is defined on line 243, $\Sigma_i$ is defined on line 102, and $\sigma(I)$ on line 104.
— Re "*terms of $\Sigma_i^c(I)$ in addition to $\Pi(\sigma(I))$ where sigma is again undefined*". The symbol $\Sigma_i^c(I)$ is defined on line 243. We will improve the wording and add a small example to better illustrate the intuitive meaning of $\Sigma_i^c(I)$.
— Re "*lines 252-256*". On lines 252 and 253, the instances of "all but one" should read "only one". We apologize for the typo.
— Re "*The best guess...along the line of play*". Your understanding is correct.
— Re "*The example also does not describe the regret update procedure*". Thanks for the feedback. We will fully work out the example in the final version using the ninth page of content.
— Re "*include a citation to Zinkevich (2008)...it would be worth pointing out*". Yes, we will. What you wrote about laminar regret vs CFR is correct, and we will make sure to point out the connection.
— Re "*Pure CFR*". Thanks for the pointer. The regret updates in Pure CFR seem very different from those of ICFR. From our understanding, Pure CFR instantiates a single external regret minimizer per decision point, while ICFR requires one internal and several external regret minimizers for each decision point. So, it is not immediately clear to us how ICFR could be a special case of Pure CFR.
— Re "*should (7) be*". Yes, good point, thanks!
— Re "*this would make it easier to read*". Thanks for the feedback, we will.
— Re "*Roughly five of this paper's eight pages are used as setup for the paper's contributions [sic]*". Since our paper combines many different concepts and tools (internal and external regret minimization, correlated equilibria, extensive-form games, etc), having a large preliminary section is unavoidable. Luckily, NeurIPS makes a ninth content page available so we will be able to expand the description of the algorithm and the example, and depending on the remaining space we will add a conclusion section accordingly.
— Re "*It is stated that ICFR is much more scalable than alternative algorithms, but there is no clear summary and accounting of the computational requirements for ICFR compared to its peers*". We never claimed that our algorithm outperforms the algorithm by Dudik and Gordon or the Ellipsoid-against-hope algorithm. However, we now offer some reasons why we believe it is reasonable to assume so. The Ellipsoid-against-hope is based on the ellipsoid algorithm and it is known in the community to be very impractical. On the other hand, the algorithm by Dudik and Gordon runs MCMC at every iteration (which is expensive) and is prone to numerical difficulties. To our knowledge, it was never tried beyond the original paper, and its implementation seems like a major effort. We argue that it is reasonable to expect that our algorithm will perform significantly better. Our algorithm is easy to implement, as it combines internal and external regret minimizers all of which can be developed and tested in isolation. It is decentralized, so each agent can be developed separately. And it relies on internal and external regret minimizers for simplex domains, for which strong practical algorithms have been developed in the past twenty years.
At any rate, that is not the point of the paper. The main point of the paper is that we give the *first* (decentralized) no-regret dynamics for EFCE and EFCCE. The algorithms by Dudik and Gordon and the Ellipsoid-against-hope algorithm do not provide no-regret dynamics. We leave the comparison of those algorithm in the context of computing one correlated equilibrium for future works in this space.

**Reviewer 3:** Thanks! If the paper is accepted, we will include in the camera-ready version a comment on the number of normal-form plans that the algorithm needs to store. The development of a procedure to avoid recording all iterates is an interesting future development. As for the scalability of our algorithm, we hope ICFR can serve as a foundation for more scalable model-free RL methods, which we believe should be a key long term goal in the multi-agent RL research agenda. Finally, we will add more details on the experimental setting in the supplementary material, so as to improve the reproducibility, and we will include a comparison on the quality of the ICFR solution vs CFR. Exploring this aspect from a theoretical perspective is another interesting research direction.

[Meta-Review · NeurIPS 2020]

All the reviewers agreed that the paper makes a significant and novel contribution. A decentralized regret minimization algorithm that converges to EFCEs is an important contribution to the field. There were some minor issues with presentation that were cleared by the author response and discussion. The connection to Pure CFR by Gibson should be referenced, and the paper should be explicitly stating the limitation that computing the EFCE requires storing every generated strategy. Please address these in the final version.